# Efficient Non-Parametric Optimizer Search for Diverse Tasks

**Ruochen Wang[1], Yuanhao Xiong[1], Minhao Cheng[2], Cho-Jui Hsieh[1]**
[1]Department of Computer Science, UCLA, [2]HKUST
ruocwang@ucla.edu    minhaocheng@ust.hk    {chohsieh, yhxiong}@cs.ucla.edu

## Abstract

Efficient and automated design of optimizers plays a crucial role in full-stack AutoML systems. However, prior methods in optimizer search are often limited by their scalability, generalization, or sample efficiency. With the goal of democratizing research and application of optimizer search, we present the first efficient, scalable and generalizable framework that can directly search on the tasks of interest. We first observe that optimizer updates are fundamentally mathematical expressions applied to the gradient. Inspired by the innate tree structure of the underlying math expressions, we re-arrange the space of optimizers into a super-tree, where each path encodes an optimizer. This way, optimizer search can be naturally formulated as a path-finding problem, allowing a variety of well-established tree traversal methods to be used as the search algorithm. We adopt an adaptation of the Monte Carlo method to tree search, equipped with rejection sampling and equivalent-form detection that leverage the characteristics of optimizer update rules to further boost the sample efficiency. We provide a diverse set of tasks to benchmark our algorithm and demonstrate that, with only 128 evaluations, the proposed framework can discover optimizers that surpass both human-designed counterparts and prior optimizer search methods. Our code is publicly available at https://github.com/ruocwang/enos.

## 1 Introductions

Motivated by a vision of democratizing machine learning, the central objective for automated machine learning (AutoML), such as automated architecture [22, 26, 29, 40, 55, 60, 62, 65] / optimizer [10, 12, 15, 17, 59, 68] / loss [36] / augmentation search [37, 39], lies in reducing the need for expert design on a diverse set of tasks. To achieve this goal, it is critical for AutoML systems to exhibit a high level of efficiency, so that they can be directly applied to a variety of tasks without consuming a humongous amount of computing resources. A widely successful example of such an effort is DARTS [40] in Neural Architecture Search (NAS), which reduces the search cost from thousands of GPU days of early RL-based algorithms to a single digit, enabling direct application of NAS systems to a wide range of tasks [32, 33, 38, 44, 50].

Inspired by the success of efficient NAS methods, we turn our attention to another important but much less studied area of AutoML - **Automated optimizer search, where an efficient, scalable and generalizable framework is still absent.** Optimizer search aims to automatically design a suitable update function that takes gradients as inputs and produces update directions for the optimizee's parameters. Pioneering work in this area, coined *Learning to Optimize (L2O)*, adopts a data-driven approach by replacing human-designed update rules with a learnable parametric function [10, 12, 17, 59]. However, parametric optimizers are fundamentally not scalable to large models or datasets, as inferring its parameters typically requires expensive meta-learning steps such as backpropagating through gradient descent [10, 15, 68]. Moreover, the learned optimizer often generalizes poorly to even minor variants of its training task (Figure 3) [10, 68]. Poor scalability and

generalization prevent L2O from being served as a general-purpose optimizer search framework that can be **directly applied to tasks of interest**.

The aforementioned limitations of parametric optimizers bring our attention to another line of method that searches over the discrete space of non-parametric update functions [1], which generally exhibit the same level of scalability and generality as human-designed optimizers [15, 51]. NOS-RL [15] extends early RL-based NAS framework [22] to optimizer search, proposing to learn a sequential controller to produce optimizer update rules according to a predefined pattern. However, NOS-RL is sample inefficient, requiring over 10k evaluations to find good candidates. More recently, AutoML-Zero [51, 69] proposes to search over the vast space of computer codes for the entire ML pipeline (including the optimizer). The excessive generality of its search space makes it even more costly to run than RL-based method. The search cost of existing non-parametric optimizer search frameworks makes them computationally prohibitive not only for practitioners to apply but also for researchers to analyze.

With the goal of democratizing research and practical applications of automated optimizer design, we introduce the first efficient, scalable, and generalizable optimizer search framework that can be directly applied to a wide range of tasks. We observe that non-parametric update rules are essentially mathematical expressions, with an innate tree structure where nodes are elementary math operators and edges represent their I/Os. Consequently, generating an update rule can be viewed as progressively appending nodes to the expression tree until it is complete. Inspired by this observation, we re-imagine the optimizer search space as a super-tree of mathematical expressions. Each leaf node on the super-tree contains an optimizer, and the path towards it represents the generation process of that optimizer's underlying expression. With the tree-structured search space, optimizer search can be naturally formulated as a path-finding problem, allowing a wide range of well-established tree-traversal methods to be used as the search algorithm. We show that a simple adaptation of Monte Carlo Sampling [30, 53], equipped with our proposed rejection sampling and equivalent-form detection, can already produce remarkable results on our search space within a fraction of budgets compared with NOS-RL ($\sim 1\%$).

We extensively evaluate the proposed framework on a diverse set of learning tasks: digit classification with MNISTNET [10], image classification with ConvNet [15], graph learning with (Cluster-)GAT [21, 28], norm-bounded adversarial attack on robustly trained models [20, 45, 46], and BERT fine-tuning on NLP datasets [34, 56]. These tasks cover both constraint and unconstrained optimizations and span over a large variety of models and datasets. Despite the simplicity, the proposed framework is able to discover update rules that surpass human-designed optimizers and prior optimizer search methods, with a budget of only 128 evaluations. We hope the proposed framework could lower the barrier of entry to practical non-parametric optimizer search, thereby providing an entry point for researchers and practitioners from ML community and beyond to study and utilize automated optimizer search systems.

## 2 Efficient, scalable and generalizable framework for optimizer search

### 2.1 Optimizer design space

**Notations and problem formulation**   Deep learning tasks are frequently expressed as optimizing a loss function $L(\cdot)$ defined over parameter domain $\theta \in \Theta$. The minimizer of $L$ can thus be obtained by $\theta^* = \arg\min_{\theta \in \Theta} L(\theta)$. For differentiable functions, a standard optimizer typically takes the form of iterative gradient descent: $\theta_{t+1} = \theta_t - \gamma * \phi(\nabla_\theta L(\theta_t))$, where $t$ is the current iteration, $\gamma$ is the learning rate and $\phi$ denote the update function. Existing optimizers primarily differ in their design of update function $\phi$; For example, vanilla gradient descent uses identity mapping $\phi(x) = x$ as the update function, whereas Adam adopts a momentum-based dynamic learning rate schema: $\phi(\nabla_\theta L(\theta_t)) = m(\nabla_\theta L(\theta_t))/\sqrt{m((\nabla_\theta L(\theta_t))^2)}$, where $m(\cdot)$ denotes the momentum function with an internal state.

The goal of optimizer search is to automatically find a suitable update function $\phi$ over some hypothesis space $\Phi$. The hypothesis spaces used in prior work can be divided into two categories: non-parametric and parametric spaces. Most human-designed optimizers belong to the first category, where the update

---

[1]Sometime it is referred to as **symbolic optimizers**, which is a somewhat inaccurate categorization as symbolic functions could also contain learnable parameters.

function $\phi$ is not trainable. Learnable optimizers, such as L2LGD2 [10] and SymbolicL2O [68], fall into the second category. Our work mainly focuses on non-parametric optimizer search, with the goal of providing an efficient, scalable and generalizable optimizer search framework that can be directly apply to various tasks.

**Optimizer update rules as expression trees**   The first step toward such a framework is to understand the structure of non-parametric optimizers. We realize that, fundamentally, optimizers are mathematical expressions consisting of elementary operators ($+$, $-$, $sign()$, $inputs$, e.t.c.). Math expressions have an inherent tree structure that preserves its order of execution, where nodes are operators and edges represent their I/Os. For instance, Diagram 1 shows the expression tree of Adam [9]:

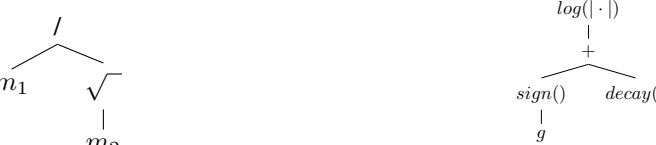

Diagram 1: Adam optimizer

Diagram 2: Our discovered Optimizer for adversarial attack

where $m_1$ and $m_2$ denote the first and second order momentum (which can also be broken down into their own expression trees).

Therefore, the generation of an update rule, as a mathematical expression, can also be conducted via top-down node selection: Take Adam as an example, we first select division ($/$) as the root node. For its left child, we pick $m_1$, which is a leaf node and thus ends the branch. For the right child, we select $\sqrt{\ }$, and subsequently pick $m_2$ to follow it. At this point, there is no empty branches left, and we obtain the complete update rule for Adam.

**A tree-structured search space**   Inspired by the completion process of update rules, we rearrange all expressions into a **super-tree**, where each leaf node contains an update rule and each path represents its completion process. The super-tree can be generated in a top-down manner: Starting from the root node with an empty update rule, we generate each of its child nodes by inserting a different operator into the update rule, and repeat this process for the generated nodes. Consequently, an optimizer can be sampled by traversing the super-tree until a leaf node is reached. Since the super-tree can grow infinitely deep, it is often desirable to restrict the tree to a predefined depth $N$, where only the paths that

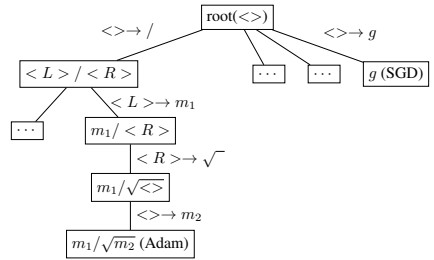

Figure 1: An illustration of traversing the super-tree to discover Adam and SGD.

can be completed within depth $N$ are included. Figure 1 provides an instantiation of our super-tree, where the paths leading to Adam and SGD optimizers are displayed as an example.

The benefit of arranging the optimizer space into a tree is two folds. Firstly, the tree-based search space is tight:

**Proposition 1** *Define the length of an update rule as the number of operators it includes, then the above tree-based search space is **tight**: a super-tree with a maximum depth of $N$ covers all update rules of length no greater than $N$.*

In a tight search space, all optimizers can be represented at the right level of complexity, allowing them to be visited by the search algorithm without exploring unnecessarily deep into the super-tree. Although tightness is a fairly obvious result for our space, it is not the case for the previous search space defined in NOS-RL, as we will explain later. Secondly, with our super-tree, optimizer search can be naturally formulated as a path-finding problem, allowing a variety of well-established tree traversal methods to be deployed as search algorithms.

**Contents** To concretize the content of the search space, we allow three types of operators in the optimizer update rule:

- a set of $p_1$ unary operators (e.g. $log(|\cdot|)$, $exp(\cdot)$, $\sqrt{|\cdot|}$, $sign(\cdot)$, $drop(\cdot)$)
- a set of $p_2$ binary operators (e.g. $+$, $-$, $\times$, $/$, $pow(\cdot, \cdot)$)
- a set of $L$ leaf values (input operators) containing gradient-based terms (e.g. $g$, $m_1$), decays (e.g. $cosine\_decay$), and constants (e.g. 1, 2)

This categorization of mathematical operators is not new, as it is also adopted in symbolic math solver [35] and NOS-RL [15].

**Comparison with NOS-RL's search space** Although both NOS-RL [15] and our framework use elementary math operators as building blocks for optimizers, they have little in common in terms of the arrangement of the search spaces. Optimizers in NOS-RL's search space are formed by a chain of predefined motifs: $b(u(I), u(I))$, where $b$, $u$, $I$ denote binary, unary and input operators. Due to the fixed structure of such motifs, NOS-RL's search space is not tight: there exist many optimizers that take extra longer sequences to express, potentially lowering their chance of being discovered by the search algorithm. For instance, Diagram 2 shows an optimizer of length 5, but it takes $(10 - 1)$ nodes (two chained motifs) to represent it under NOS-RL's arrangement; Moreover, NOS-RL's search space also requires extra bypass operators (e.g. $u(x) = x$ and $b(x, y) = x$) to cover even human-design optimizers such as Adam and PGD, further increasing the complexity. In contrast, our representation of optimizers is directly inspired by the innate structure of its underlying mathematical expressions, resulting in a tight tree-based search space. In our search space, optimizer search can be naturally formulated as a form of top-down path-finding problem. In the next sections, we will detail our choice of algorithms for traversing the super-tree, as well as several techniques that leverage the characteristics of optimizer update rules to boost the sample efficiency.

## 2.2 Monte Carlo Sampling for tree traversal

We adopt a simple adaptation of Monte Carlo Sampling to tree traversal [30, 53, 70] (MCT) as the search algorithm. The idea is to assign scores to the nodes in the super-tree (Figure 1), and use these scores to guide the tree traversal. We define the score of a node $v$ as a Monte Carlo estimation over unrolling steps from $v$: If $v$ is an internal node, we randomly generate a set of unrolled paths from $v$ to the corresponding leaf nodes, and take the average score of the resulting optimizers as the score for $v$; If $v$ is a leaf node, we set its score to 0 as it cannot be expanded. The search can thus be conducted as follows: 1). Starting from the root node $v^{(0)}$ at level 0, we generate all child nodes $\{v^{(1)}\}$ of $v^{(0)}$ by inserting each operator from the candidate pool to the update rule in $v^{(0)}$; 2). From there, we select the child node $v^{*(1)}$ with the highest MC score to expand, and move on to the next level; 3). The process is repeated until a predefined maximum search level is reached. Algorithm 1 in the Appendix provides a detailed summary of the complete search process.

Directly applying the MCT algorithm to optimizer search would not perform well under limited search budgets, due to two unique characteristics of optimizer update rules that challenge the sample efficiency of the Monte Carlo estimates. Firstly, the majority of mathematical expressions, when deployed as optimizer update rules, perform poorly or even would not converge. This is usually not the case for other AutoML tasks such as neural architecture search, as most networks in the search space perform reasonably well. The large body of poor-performing optimizers not only consumes precious search budget, but also causes the MC estimation to be unstable. Secondly, there exists many mathematical redundancies in the expression space, for example: $sign(sign(sign(x)))$ can be reduced to $sign(x)$, and $\frac{m_1 + \sqrt{m_2}}{\sqrt{m_2}}$ is equivalent to $\frac{m1}{\sqrt{m2}} + 1$. Identifying and eliminating these redundancies would not only save budget, but also prevent the sampling distribution from biasing toward mathematically simple and shallow update rules. To address these issues and further boost the sample efficiency, we propose two sets of techniques - rejection sampling and equivalent-form detection. When combined with these techniques, the simple MCT algorithm becomes particularly effective for the optimizer search task. We will discuss them in detail in the following sections.

## 2.3 Rejection sampling

**Eliminating poor optimizers with a train-free task-agnostic test**   Inspired by the characteristics of optimizer update rules, we develop a train-free task-agnostic test to eliminate poor optimizers without evaluating them. We propose a necessary condition for a valid optimizer: it must produce an acute angle with steepest descent direction (i.e. gradients). We check the validity of optimizers against this condition and only evaluate those that pass the test. For the test to be task-agnostic, we feed the optimizer with a batch of random Gaussian vectors in place of actual gradients. Formally, the descent test can be written as:

$$E_{u \sim \mathcal{N}(0,1)} \big[ \cos(\phi(\nabla_\theta \mathcal{L}), \nabla_\theta \mathcal{L}) \big] > \lambda_d$$

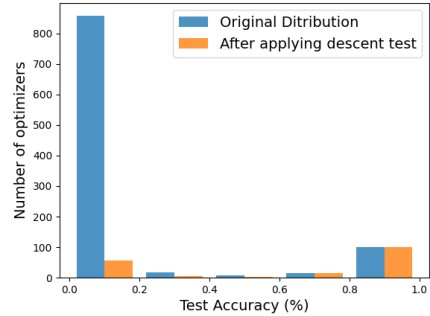

Figure 2: Performance distribution of optimizers after applying descent test, under $\lambda_d = 0.15$ and a batch size of 25.

where $\lambda_d$ is a predefined threshold. Although our descent test is by-no-mean comprehensive, it can effectively rule out a large chunk of poor optimizers with negligible false-negative rates, as demonstrated in Figure 2.

**Reducing the variance of MC estimates via score thresholding**   After applying the descent test, there still remains a non-negligible portion of poor optimizers. When sampled during the unrolling step, these optimizers would drastically lower the Monte Carlo score of the stem node, causing the MC estimation to exhibit high variance and thus become unreliable. This adverse effect is especially severe under the efficient setting when the sample size is small. Therefore, we propose to simply reject candidates with scores lower than a predefined threshold, thereby removing them from the MC scores of the corresponding stem nodes.

## 2.4   Detecting and handling redundancies in mathematically equivalent forms

**On-the-fly constraint tree-traversal for redundant path pruning**   One benefit of formulating the search problem as top-down path-finding is that we can easily apply constraints on-the-fly to eliminate undesirable branches - those that lead to mathematically redundant expressions in our case. We identify three main categories of such constraints:

- child operator that nullifies its parent's operator. e.g. $-(-x) = x$, $ln(e^x) = x$
- child operator that is redundant under its parent. e.g. $clip(clip(x)) = clip(x)$
- sequence of operators that reduces to a constant in the search space. e.g. $\sqrt{|sign(x)|} = 1$

The complete sets of constraints we used can be found in the Appendix. Enforcing these constraints during the traversal can effectively trim down the search tree, allowing the algorithm to explore branches that lead to more diverse and complex expressions.

**Hashing mathematically equivalent expressions**   Besides enforcing constraints during the traversal, it is also important to detect mathematically equivalent optimizers to avoid duplicated evaluations. One can always apply off-the-shelf symbolic solvers to identify the equivalence of two expressions, $\phi$ and $\phi'$, by checking if $(\phi - \phi')$ can be reduced to 0. However, it could become extremely slow as the pool of evaluated optimizers $\{\phi_i\}_1^N$ gets larger and larger, since we need to solve $N$ pair of equations every time a new update rule is sampled. Instead, we apply hashing to efficiently query the evaluated candidate pool for mathematically equivalent optimizers. Concretely, we assign each optimizer a hash code, obtained by feeding a fixed probing vector as input to the optimizer and recording its output. The probing vector is pre-sampled from Gaussian distribution. When a newly sampled optimizer arrives, we only need to compare its code with the hash table to check the existence of its equivalent form. Empirically, it is much faster to run the proposed hashing-based checker than symbolic solvers.

## 3   Discussions and relationship to prior work

**Automated optimizer design**   Optimization plays a crucial role in training deep learning models. Generally, there does not exist one optimizer that aces all scenarios, as different tasks (dataset,

architecture, loss, parameterization, e.t.c.) might favor different optimization methods [2, 10]. The demand for task-specific optimizers stimulates research interest in developing automated systems for optimizer design [10, 12, 15, 17, 41, 51, 59, 66, 68]. Early work adopts a data-driven method by modeling the optimizer update with a parametric function [10, 12, 59]. L2LGD2 [10] deploys an LSTM model as the update function that takes historical gradients as input and produces the update direction. However, parametric optimizer search methods are fundamentally limited by its scalability, as inferring its parameters requires expensive meta-learning steps such as back-propagation through optimization [10, 59, 68]. Although SymbolicL2O [68] improves the scalability of the learned LSTM optimizer by distilling it into a lightly-parameterized symbolic optimizer, it still requires a pretrained LSTM model to begin with. Instead of learning a parametric optimizer, NOS-RL [15] directly searches over a discrete space of non-parametric update functions comprised of mathematical operators. It extends early RL-based NAS method [22] to optimizer search, by training a sequential controller to produce the optimizer update rule according to a predefined pattern. However, similar to its NAS counterpart, NOS-RL is also computationally expensive, requiring over 10k evaluations to find good candidates.

**Symbolic optimization and differential program synthesis** Symbolic optimization (SO) [1, 13, 53, 61, 63, 64] aims at optimizing an objective over a symbolic hypothesis space of functions (or more broadly, programs). One line of work attempts to recover the unknown equation from its generated data, with great potential in automating scientific discoveries [54, 64]. Another line of methods aims at finding a more interpretable and generalizable symbolic model to replace the black-box neural networks [53, 61, 63]; Applications that witnessed some success include learning symbolic policy networks for RL [63] and sequential classification models [53, 61]. The latter is often studied under the concept of Program Synthesis [24], where a model is extended to include programmatic rules such as if-else clause, indexing, e.t.c. SO is closely connected to AutoML at a high level, as both fields frame their problems as discrete optimization. Indeed, many existing optimizer search methods can find their counterparts in symbolic optimization. Our method is also inspired by the rich body of literature in deep symbolic mathematics and program synthesis, which also explores tree-based expression spaces for differential equations and programs [30, 35, 53, 61, 64]. However, due to significant differences in taskonomy, SO and AutoML methods are often developed separately, converging into different branches of techniques. Symbolic optimization often studies tasks where candidates are cheap to evaluate but finding the global optimal is desired [1, 54, 64]; As a result, sample efficiency is often not the primary concern. Much to the opposite, in AutoML tasks, candidate evaluations are extremely expensive; Therefore, it is more beneficial to identify a good-enough candidate within a limited amount of budget.

## 4 Empirical evaluations on a diverse set of tasks

We extensively evaluate the proposed framework on a suite of tasks, covering a variety of models and datasets. On standard benchmark tasks for optimizer search, our method is able to discover optimizers that outperform its human-designed and automatically searched counterparts. In addition, we also show that the proposed framework enables automated optimizer design for many other popular learning tasks, such as adversarial attack, GNN training, and BERT finetuning. Due to the space limits, we will include detailed descriptions, search settings, and discovered optimizers for each task in the Appendix.

### 4.1 General setting

**MCT algorithm** Across all experiments, we limit the maximum level of MCT traversal to 4, and set the number of Monte Carlo samples to 32 (a multiple of 8 for parallelism on 8-GPU servers) for each level. This amounts to a fixed total budget of 128 evaluations. The maximum depth for the super-tree is set to 10, which already covers many top-performing optimizers for various tasks. We use a similar set of elementary operations as NOS-RL to build the optimizers, with only minor adjustments for some tasks (see Appendix for more details).

**Optimizer evaluation** We follow the default settings and hyperparameters for each task, and only swap out the optimizer; This potentially puts our algorithm at a disadvantage, as the hyperparameters are usually tuned around the default optimizers. Before optimizer evaluation, we perform grid search

on a small proxy task (fewer steps) to find a proper learning rate. During the grid search, we also aggressively terminate optimizers if their performance falls under a certain threshold. Since early stopped optimizers consume fewer resources than a full evaluation, we do not count them into the budget (number of evaluations).

## 4.2   Hand-written digit classification

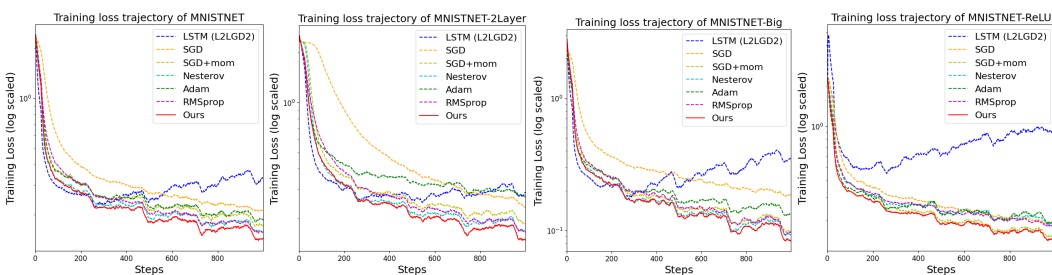

Figure 3: Training loss trajectory on hand-written digit classification task (log scaled). Each optimizer is evaluated for 4 random seeds. Our method is marked in red.

We first compare our method with the LSTM-based optimizer (L2LGD2) on hand-written digit classification. Following L2LGD2 [10], the goal is to minimize the cumulative training loss of a single-hidden-layer MLP with Sigmoid activation (MNISTNET) on the MNIST dataset; The search is conducted on MNISTNET for 100 steps with a batch size of 128, and the discovered optimizers are subsequently transferred to three variants of MNISTNET with different activations (MNISTNET-ReLU), number of hidden layers (MNISTNET-2Layer), and dimensions (MNISTNET-Big). Under this setting, our method finishes in 0.92h on RTX 2080ti, much faster than L2LGD2 (2.62h).

As shown in Figure 3, our discovered optimizer achieves the lowest training loss under both direct search and transfer settings. Notable, the LSTM-based parametric update function indeed converges faster when the number of steps is close to the search phase (black-dotted vertical line on Figure 3). However, it extrapolates poorly to longer trajectories. As the training proceeds, all other non-parametric optimizers eventually catch-up, achieving much lower training loss. Moreover, LSTM-based optimizer also generalizes poorly to other model variants (most noticeably MNISTNET-ReLU), revealing its tendency to overfit the search task.

## 4.3   Image classification with ConvNet

We proceed to evaluate our method on the CIFAR-10 [8] classification task proposed in NOS-RL [15]. The model of choice is a 2-layer ConvNet. Each layer of this network contains a 32-filter 3x3 convolution with ReLU activation and batch normalization. Following NOS-RL's setting, for every optimizer, the best learning rate is searched over a grid of $\{1e^{-5}, 1e^{-4}, 1e^{-3}, 1e^{-2}, 1e^{-1}, 1\}$ with 1 epoch of training, and the discovered learning rate is subsequently used to train the model for a longer period of time (5 epochs). Since NOS-RL's implementation is not open-sourced, we reproduce and compare with the two families of discovered optimizers described in NOS-RL paper: AddSign and PowSign.

The results are summarized in Table 1. For NOS-RL, we display the performance of the top 4 variants of PowSign and AddSign, which are obtained after training the controller for over 10k evaluations (Figure 4 in the NOS-RL paper [15]). With only a fraction ($\sim$1%) of the search budget, our framework is able to discover optimizers that reach a test accuracy of 77.02%, topping both PowSign and AddSign optimizers and also human-designed ones by a sizable margin. The sheer reduction in search cost and the improvement in search performance evince the efficiency and effectiveness of the proposed framework for discovering better optimizers.

## 4.4   Adversarial attack

Next, we apply our framework to discover optimizers for constraint optimization. We select adversarial attack, which aims at finding norm-bounded perturbations in the input space that alter the model's predictions. The de facto optimizer used in adversarial attack is Projected Gradient Descent

Table 1: Performance of automated search algorithms on CIFAR-10.

| Optimizer | Test Accuracy (%) | Search Method | Search Budget (#evaluations) |
|---|---|---|---|
| SGD | 70.99% ± 2.12 | manual | - |
| SGD + Momentum | 74.12% ± 0.44 | manual | - |
| Nesterov | 74.15% ± 0.52 | manual | - |
| Adam | 73.42% ± 0.56 | manual | - |
| RMSprop | 71.42% ± 1.42 | manual | - |
| PowSign-ld | 75.48% ± 0.45 | RL on hand-crafted patterns | >10,000 |
| PowSign-cd | 76.21% ± 0.16 | RL on hand-crafted patterns | >10,000 |
| AddSign-ld | 75.54% ± 0.39 | RL on hand-crafted pattern space | >10,000 |
| AddSign-cd | 76.07% ± 0.59 | RL on hand-crafted pattern space | >10,000 |
| Ours | **77.02% ± 0.19** | MCT on super-tree space | **128** |

(PGD) [20]. We consider the most popular $l_\infty$-norm setting. For $l_\infty$-norm bounded attack, PGD takes the form of: $x = Proj_{B_\epsilon^\infty(x_o)}(x + \gamma sign(\nabla_x L(x))))$, where $B_\epsilon^\infty(x_o)$ represents a $\epsilon$ ball around the original image $x_o$ w.r.t. $l_\infty$-norm. The models of choice come from the AutoAttack library [46], which holds a leaderboard of top defense methods. Following their settings, we set $\epsilon = 8/255$, and run each optimizer once for 100 steps on every image from the test split [46].

On this task, we mainly search for the update rule inside the projection operator (e.g. $sign()$ for PGD). The search is conducted on the pre-trained Carmon2019 model [27], and the proposed optimizer is subsequently evaluated on other top defense methods for WideResNet [14] (WRN-<depth>-<width>) and ResNet [11] (RN-<depth>). As shown in Table 2, our discovered optimizer consistently outperforms PGD by a sizable margin. Surprisingly, we found that the algorithm tends to pick $log(|\cdot|)$ rather than $sign(\cdot)$ as the first operator, resulting in many log-based optimizers that surpass sign-based PGD.

Table 2: Attack success rate of different optimizers on top defense methods on CIFAR-10.

| Defense Models | PGD | APGD | Ours |
|---|---|---|---|
| Carmon2019 (WRN-28-10) [27] | 37.83% | 38.22% | **38.35%** |
| Gowal2020‡ (WRN-70-16) [48] | 31.10% | **32.00%** | **32.00%** |
| Gowal2020‡ (WRN-34-20) [48] | 40.05% | 40.46% | **40.50%** |
| Gowal2020‡ (WRN-28-10) [48] | 33.65% | 34.33% | **34.34%** |
| Sehwag2020‡ (WRN-28-10) [52] | 40.00% | 40.43% | **40.46%** |
| Wu2020‡ (WRN-28-10) [58] | 36.41% | 36.70% | **36.78%** |
| Wang2020‡ (WRN-28-10) [42] | 37.78% | 38.16% | **38.27%** |
| Engstrom2019 (RN-50) [31] | 47.76% | 48.25% | **48.32%** |
| Wong2020Fast (RN-18) [57] | 53.69% | 54.11% | **54.19%** |

‡ Methods that explore extra data during robust training.

In addition to PGD, we also compare our log-based optimizer with the best handcrafted and tuned optimizer for adversarial attack: Adaptive PGD (APGD) [46]; The design of APGD is packed with domain expertise: it combines a well-tuned momentum update rule with a conditional learning rate decay based on a handcrafted schedule and sophisticated decay conditions (see Appendix for details). However, the performance of our automatically discovered optimizer rivals APGD across various defense methods, despite of having a much simpler form (see Appendix for details). This result demonstrates the potential of applying our framework to reduce the need of human expertise in designing optimizers for diverse tasks.

## 4.5 Node classification on graphs

We next test our framework for optimizing graph neural networks to classify nodes on graphs. The model of interest is Graph Attention Network (GAT) [21], one of the most widely used architectures in graph learning tasks. We compare our method against Adam [9] - the standard optimizer for optimizing GATs - on five commonly used graph datasets: OGBN-Product [49], Cora [4], Citeseer [3], PubMed [6], and PPI [18]. Among them, OGBN-Product is the largest in scale, consisting of 2,449,029 nodes. Since standard GATs cannot scale to this dataset, we instead adopt an adaptation of cluster-GCN [28] to GAT as the testbed, termed Cluster-GAT. Cluster-GAT trains standard GAT on smaller partitions of the original graph, thereby allow-

Table 3: Performance of our discovered optimizers against Adam on GATs on five commonly used Graph datasets of diverse size. Results that use the same GAT implementations are grouped together.

| Dataset | Adam | Ours |
|---|---|---|
| Products | 77.49% ± 0.56† | **80.15% ± 0.16** |
| Cora | 84.72% ± 0.32 | **85.20% ± 0.19** |
| Citeseer | 71.70% ± 1.03 | **73.10% ± 0.43** |
| PubMed | 78.20% ± 0.22 | **79.25% ± 0.70** |
| PPI | 97.53% ± 0.45‡ | **98.13% ± 0.10‡** |

† Our reproduced accuracy using ogbn-leaderboard's implementation is lower than the displayed number (79.23% ± 0.78).

‡ F1 Score

ing the model to be applied to large-scale graphs. We refer the reader to the Appendix for detailed descriptions of all GAT implementations and experimental setups.

The results are summarized in Table 3. On all datasets, our search algorithm is able to discover optimizers that outperform Adam. An interesting observation is that the top-performing optimizers discovered for this task almost always contain $sign(\cdot)$ operators, revealing the potential of adopting sign-based optimizers to improve the training of graph neural networks.

### 4.6 BERT fine-tuning on NLP datasets

We also evaluate the proposed framework on BERT finetuning task on GLUE benchmark [25]. For this task, we follow all configurations of the Hugging-Face [56] implementations: we finetune a pretrained BERT (base cased) model for 3 epochs on Cola [43], STS-B [16] and RTE [7] dataset, and 5 epochs on MRPC [5] and WNLI [25] dataset. The batch size is set to 32. We compare our discovered optimizers with the default AdamW [19]. As shown in Table 4, our automatically discovered optimizers outperform AdamW on all datasets.

Table 4: Performance of our discovered optimizers for BERT finetuning on GLUE tasks.

| Dataset | AdamW | Ours |
|---|---|---|
| Cola | $59.56 \pm 2.04^{\star}$ | $\mathbf{60.89 \pm 1.33}^{\star}$ |
| MRPC | $82.84 \pm 0.57^{\ddagger}$ | $\mathbf{86.64 \pm 0.94}^{\ddagger}$ |
| STS-B | $87.80 \pm 1.14^{\dagger}$ | $\mathbf{88.91 \pm 0.30}^{\dagger}$ |
| RTE | $65.97 \pm 1.56^{\ddagger}$ | $\mathbf{68.50 \pm 1.93}^{\ddagger}$ |
| WNLI | $53.17 \pm 5.49^{\ddagger}$ | $\mathbf{56.34 \pm 0.00}^{\ddagger}$ |

$^{\star}$ Mathews Correlation.
$^{\dagger}$ Spearman Correlation.
$^{\ddagger}$ Accuracy (%).

## 5 Ablation study

In this section, we ablate the proposed framework using the MNISTNET task. All experiments are repeated over 4 random seeds to account for randomness in the search phase.

**Random search baseline**  We study the effectiveness of our MCT algorithm alone by comparing it with random sampling. Concretely, instead of traversing the tree based on MC scores, we randomly generate all optimizers from the root. Everything else in our framework remains unchanged, including our rejection sampling and equivalent-

Table 5: Performance comparison of Random Search and MCT.

| Method | Training Loss (sum) | Test Accuracy |
|---|---|---|
| Random | $60.25 \pm 2.99$ | $88.02\% \pm 1.53$ |
| MCT | $\mathbf{59.25 \pm 2.50}$ | $\mathbf{89.43\% \pm 0.85}$ |

form detection techniques. This is equivalent to Random Search on our search space. As shown in Table 5, MCT algorithm outperforms Random Search baseline by a sizable margin, showing that the Monte Carlo node scoring schema can indeed guide the traversal towards promising branches of the tree.

**Score thresholding**  As discussed in prior sections, score thresholding is important to the performance of the MCT algorithm. To verify this, we ablate this technique by disabling it in our framework while keeping everything else the same. Without score thresholding, the cumulative training loss of the proposed optimizers raises from $59.25 \pm 2.50$ to $60.25 \pm 2.87$, similar to that of random search.

## 6 Conclusion

Despite the recent advancement of practical AutoML systems in automatizing the design of architectures, data augmentation policies, and hyperparameters, progress in automated discovery of optimizers is still inadequate due to the limitations of prior methods in terms of 1). efficiency, 2). generalization, and 3). scalability. In this paper, we introduce the first optimizer search framework that meets all these criteria, allowing it to be directly applied to the tasks of interest. The proposed framework demonstrates promising results across a variety of tasks, from image classification, adversarial attack, to graph learning and BERT finetuning. Our method by-no-mean solves the optimizer search problem, as there is plenty of room for improvement on the algorithm and search space; Rather, our goal is to open up a new possibility for future development in non-parametric optimizer search methods. We hope the proposed framework could democratize research and applications of automated optimizer search, and stimulate interest among researchers and practitioners.

# 7 Limitations

Our view of this work is as a starting point of an efficient, scalable, and generalizable framework for optimizer search. And we expect plenty of room for improvement for future works. For instance, we identify the following concrete limitations of the method:

1. We use pre-computed momentum terms as input to our search space. This is a practice borrowed from NOS-RL. Adding commonly used terms ease the job of the search algorithm because it does not have to rediscover them from scratch every time. However, searching for novel momentum update rules could potentially help to find even stronger optimizers. In principle, our framework allows it: one can do this by inserting an operator with its own internal state. This would serve as a direction for future explorations.

2. Identifying proper hyperparameters for an optimizer is essentially for evaluation. In the current work, we use a simple grid search to discover the best learning rate for an optimizer. While it works fine for our tasks, this could potentially be suboptimal as it might underestimate some optimizers. Leveraging advanced fast HPO during the search phase could be another direction to explore.

3. Although our framework is 100x faster than the comparable method (NOS-RL), it still requires 128 evaluations in the search phase. These evaluations can be largely parallelized. But potentially, the efficiency can be improved further with better search algorithms, more train-free tests, knowledge transfer, e.t.c.

## Reproducibility & ethics statements

**Reproducibility** We have specified the setup for ours experiments in the main paper and Appendix, including settings for each task and hyperparameters for our method. The code and the optimizers found by our methods will also be published on Github upon acceptance to encourage future development. Before then, a copy of our code is included in the supplementary material for reference.

**Ethics** We are not aware of any potential ethical concerns regarding our work.

## Acknowledgments and Disclosure of Funding

This work is supported in part by NSF under IIS-2008173, IIS-2048280, an Okawa research grant and a Google research scholar award.

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
