# A Appendix

## A.1 Pseudocode for our search algorithm

Our framework follows a standard search pipeline:

1. Candidate proposal: the search algorithm samples an optimizer from the search space.

2. Candidate evaluation: The optimizer is evaluated from scratch, by using it to optimize a model and obtaining the performance of the model.

3. Search: The optimizer score is used to guide the search algorithm to propose new optimizers.

4. Loop: Repeat step 1 - 3 until a predefined search budget is reached.

5. Finally, the Top optimizers visited by the search algorithm will be returned.

This procedure is commonly used in other AutoML domains, such as Neural Architecture Search [47, 67] and Hyperparameter Optimization [23]. Algorithm 1 and 2 summarize the complete search process.

---

**Algorithm 1:** MCT algorithm

---

1 **Input:** Candidate set $\mathcal{A}$, constraints $\mathcal{C}$, operator set $\mathcal{O}$, maximum super-tree depth $D$, maximum traversal level $L$, MC sample size $M$ for each level, score threshold $\rho$, proposal size $K$.

2 **Main search:**

3 **for** *level in* $1$ *to* $L$ **do**

4      current node $v_c$ = root node ;        // root node hosts an empty update rule

5      score dict $S = \emptyset$ ;       // scores of optimizers generated from stem nodes

6      number of samples $m = 1$

7      **while** $m <= M$ **do**

8          $u$ = randomly pick a child of $v_c$, by inserting an operator $o \in \mathcal{O}$ to $v_c$ under constraint $C$;

9          $\phi = unroll(u, \mathcal{C}, \mathcal{O}, D)$;

10          **if** *not descent_test($\phi$)* **then**

11             continue;

12          $s_\phi = evaluate(\phi)$ ;       // the score for early stopped $\phi$ are set to $\rho$

13          **if** $s_\phi > \rho$ **then**

14             register $(u, s_\phi)$ in $\mathcal{S}$;

15             register $(\phi, s_\phi)$ in $\mathcal{A}$;

16             $m$ += 1;

17      **for** *each node $u$ in $\mathcal{S}$* **do**

18          **if** *$u$ is a non-leaf node* **then**

19             compute the average score of $u$;

20          **else**

21             set the score of $u$ to 0;

22      $v_c$ = node with the best score as computed above ;      // move on to the next level

23 **return:** $Top_K(\mathcal{A})$;

---

# B Set of operators used for constructing the search space

Inspired by NOS-RL, we adopt the following set of mathematical operators in our experiments:

- Unary operators: $-(\cdot)$, $exp(\cdot)$, $log(|\cdot|)$, $\sqrt{|\cdot|}$, $clip_{0.003}(\cdot)$, $drop_{0.1}(\cdot)$, $sign(\cdot)$
- Binary operators: $+$, $-$, $\times$, $/$, $pow(\cdot, \cdot)$

---
**Algorithm 2:** Pseudocode for the unrolling step
---
1 **Input:** Stem node $v$, constraint $\mathcal{C}$, operator set $\mathcal{O}$, maximum super-tree depth $D$

2 **Unroll:**
3 set current node $v_c = v$

4 **while** *True* **do**
5 $\quad$ $v_c$ = randomly pick a child of $v_c$, by inserting an operator $o \in \mathcal{O}$ to $v_c$ under constraint $C$;
6 $\quad$ **if** $\phi_{v_c}$ *is a complete update rule* **then**
7 $\quad\quad$ break;
8 $\quad$ **else if** *length*($\phi_{v_c}$) == $D$ **then**
9 $\quad\quad$ $v_c = v$ ; $\qquad\qquad\qquad\qquad\qquad\qquad$ // restart unrolling
10 $\quad\quad$ continue;

11 **return:** $\phi_{v_c}$
---

- Input Operators: $g$, $g^2$, $g^3$, $m_1$, $m_2$, $m_3$, $sign(g)$, $sign(m_1)$, $Adam$, $RMSprop$, 1, 2, $ld$, $cd$, $rd$

Here, $m_1$, $m_2$, $m_3$ denote the first, second and third order momentum respectively, and $ld$, $cd$, $rd$ denote linear decay, cosine decay, and restart decay [15]:

$$\textbf{linear decay}: \quad 1 - \frac{t}{T}$$

$$\textbf{cosine decay}: \quad 0.5 * (1 + \cos(2\pi n \frac{t}{T}))$$

$$\textbf{restart decay}: \quad 0.5 * (1 + \cos(\pi \frac{(tn)\%T}{T}))$$

where t and T are the current and maximum step. Following NOS-RL, we use $n = 0.5$ for cosine decay and $n = 20$ for restart decay. We set the bound for $clip$ operator to 0.003, and the dropout ratio to 0.1 for $drop$ operator. Note that one can always include more options of these values by adding new operator variants to the space (e.g. $drop_{0.3}()$ with dropout ratio set to 0.3). For all input operators, we use their default PyTorch implementations and hyper-parameters. The only exception is the learning rates for Adam and RMSprop. We found that under the default learning rate, the norm of Adam and RMSprop is sometimes quite small compared with other operands such as $sign(g)$, making them potentially less effective as a submodule of some optimizers. Therefore, we raise their default learning rate by $3\times$ in our experiment. Note that we make one minor adjustment to the set for the ConvNet task: $g^3$, $m_3$, $Adam$, and $RMSprop$ are removed as they rarely show up on the tops optimizers.

Our set of operators is a subset of the full operator set presented in Section 4.1 of the NOS-RL paper. However, note that NOS-RL also uses much smaller subsets rather than the full set to conduct the search. We refer the readers to "Further discussions on NOS-RL baseline" in Appendix D.3 for more details.

**Suggestions on constructing operator sets for future tasks** Our provided set of operators and heuristics are quite generic. Empirically, we found that the operator set is good enough for constructing high-performant optimizers for all tasks used in this paper, including MNIST, CNN training, BERT finetuning, GNN training, and adversarial attacks. And we use the same on-the-fly constraints, train-free test, and score threshold for all experiments.

For future tasks, we recommend starting with the provided set and incorporating new operators when necessary. For inspiration on what to add, the user might look into 1). what extra mathematical operations are used in existing optimizers for this task 2). the nature of the task (e.g. some might require max or sinusoid functions).

With an augmented operator set, other components in our algorithm can largely remain the same. 1). our train-free test, score thresholding, and math equivalent detection are generic and independent of the operator set 2). It could be beneficial to augment the on-the-fly constraints slightly when the

new operators are added. It is because the on-the-fly constraints are mainly used for eliminating mathematical redundancies, and new redundancies might surface when new operators are included. But this should be an extra gain. We encourage the users to follow our general guidelines in Section 2.4 for spotting these redundancies.

## C    Choice of hyperparamters for the search algorithm

We will elaborate more on our choice of hyperparameters for the search phase. Note that our search hyperparameters are held fixed throughout all our experiments, across all the tasks.

**Score threshold**    The purpose of score thresholding is mainly to reject apparently poor optimizers - those that lead to near-random-guess accuracy or exploding loss. So we simply set it (20%) to be slightly higher than random guessing accuracy and used it for all our experiments.

**Number of samples in Monte Carlo estimation**    We set and fix the number of total samples in Monte Carlo estimation to 32 - a multiple of 8 for parallelization. If this number is too small, then the MC scores for each branch would be of high variance; Setting this number too big does not hurt performance, but also might not be necessary as it consumes extra budget.

**Super Tree Depth**    We choose ten because we found that most top optimizers are within this range. We also experiment with increased depth (15, 20) but only find marginal gains sometimes. The discovered longer optimizers are also less interpretable.

**The number of traversal levels**    We use a fixed number to fix the same budget for all experiments (recall that num_mc_samples * num_levels = total budget) (line 267). But in practice, this hyperparameter needs not to be preset or tuned; One can just progressively increase it and stop when there is no further gain in terms of the proposed optimizer's performance.

**Learning rates for optimizer evaluation**    We perform simple grid search to identify a suitable learning rate for each optimizer. We use one main grid for our experiments, **without** tuning it just to get more favorable results. However, we adjust the grid slightly for different tasks for the following reasons:

1. We use $\{1e^{-5}, 1e^{-4}, 1e^{-3}, 1e^{-2}, 1e^{-1}, 1\}$ for ConvNet task proposed in NOS-RL. This grid is too sparse, but it is adopted in NOS-RL (Sec 4.3). So we have to use the same grid for a fair comparison with NOS-RL.

2. For all other tasks, we use this grid: $\{0.0001, 0.0003, 0.0006, 0.001, 0.003, 0.006, 0.01, 0.03$ $, 0.06, 0.1, 0.3, 1.0\}$. One can always use the full grid. However, this grid is sometimes unnecessarily long for some tasks - the first or the last few rates are never selected. Due to the sheer size of experiments in this paper and limited resources, we often shrink this grid mildly to speed up the runs a little. The exact truncated grids for each task are reported in the following section.

3. We always want to ensure that the grid covers the provided learning rate of our baseline optimizers for a fair comparison. For the BERT task, since the baseline (AdamW)'s learning rate is set to $2e^{-5}$ (outside this grid), we extend our grid to cover it, avoiding underestimating AdamW.

## D    More details on experimental settings

### D.1    General settings for our search algorithm

**Search configurations**    For all experiment, we allow 4 levels of traversal and set the number of Monte Carlo samples for each level to 32. This amounts to a total budget of 128 evaluations. The maximum depth for the super-tree is set to 10. The evaluation of Monte Carlo samples for each level of traversal are completely independent, and therefore can be easily parallelized onto multiple GPUs. As mentioned in the main text, we also apply score thresholding during the Monte Carlo estimation. We use a universal threshold of 10 for losses, 20% for accuracy and correlations. After the search phase, top 5 optimizers are usually proposed for further evaluations.

**Early stopping**   We also early stop poor optimizers to speedup the search process. We use the following standard procedure for deciding whether to terminate the training of an optimizer: If the search signal is training loss, we track if the moving average of the training loss keeps increasing for certain amount of consecutive steps. If the search signal is accuracy or correlations, we check if the accuracy fails to reach the score threshold after $10\%$ of training.

**Constraints**   We use the following constraints during tree traversal: 1). $log(exp(\cdot))$ and $-(-(\cdot))$ are prohibited. 2). $sign(\cdot)$ must not be followed by $sign(\cdot)$, $sign(m_1)$, $sign(g)$, $clip_{0.003}(\cdot)$, 1, 2, $ld$, $cd$, and $rd$. 3). $\sqrt{|\cdot|}$ must not be followed by $sign$ and 1. 4). $clip_{0.003}(\cdot)$ must not be followed by $clip_{0.003}(\cdot)$, 1, 2, $ld$, $cd$, and $rd$.

## D.2   Hand-digit classification with MNISTNET

Table 6: Performance of different optimizers on MNISTNET models.

| Method | MNISTNET | | MNISTNET-2Layer | | MNISTNET-Big | | MNISTNET-ReLU | |
|---|---|---|---|---|---|---|---|---|
| | Train Loss (Sum) | Test Acc | Train Loss (Sum) | Test Acc | Train Loss (Sum) | Test Acc | Train Loss (Sum) | Test Acc |
| SGD | $364.96 \pm 3.32$ | $93.09\% \pm 0.17$ | $638.23 \pm 12.91$ | $92.27\% \pm 0.44$ | $334.72 \pm 1.90$ | $93.88\% \pm 0.08$ | $317.33 \pm 7.47$ | $93.56\% \pm 0.37$ |
| SGD + Mom | $276.26 \pm 10.78$ | $93.07\% \pm 0.46$ | $358.61 \pm 8.96$ | $93.05\% \pm 0.32$ | $207.54 \pm 5.12$ | $95.29\% \pm 0.25$ | $265.15 \pm 9.58$ | $94.03\% \pm 0.53$ |
| Nesterov | $248.96 \pm 6.51$ | $93.53\% \pm 0.32$ | $317.86 \pm 6.38$ | $93.32\% \pm 0.32$ | $192.03 \pm 13.13$ | $95.35\% \pm 0.31$ | $283.50 \pm 41.82$ | $92.95\% \pm 0.83$ |
| Adam | $327.15 \pm 11.55$ | $91.54\% \pm 0.53$ | $403.07 \pm 31.20$ | $90.69\% \pm 0.54$ | $219.25 \pm 4.43$ | $94.29\% \pm 0.33$ | $273.02 \pm 15.47$ | $92.56\% \pm 0.72$ |
| RMSprop | $269.48 \pm 5.74$ | $93.72\% \pm 0.17$ | $336.99 \pm 13.33$ | $93.44\% \pm 0.37$ | $230.69 \pm 4.30$ | $95.01\% \pm 0.20$ | $280.28 \pm 11.59$ | $93.61\% \pm 0.33$ |
| L2LGD2 | $300.94 \pm 12.49$ | $90.63\% \pm 0.14$ | $338.18 \pm 11.69$ | $90.11\% \pm 0.30$ | $286.63 \pm 8.33$ | $90.94\% \pm 0.32$ | $791.35 \pm 55.13$ | $84.24\% \pm 1.49$ |
| Ours | $\mathbf{237.76 \pm 5.34}$ | $\mathbf{93.86\% \pm 0.23}$ | $\mathbf{291.90 \pm 7.89}$ | $\mathbf{93.75\% \pm 0.38}$ | $\mathbf{186.17 \pm 6.68}$ | $\mathbf{95.42\% \pm 0.16}$ | $\mathbf{238.19 \pm 8.37}$ | $\mathbf{94.29\% \pm 0.30}$ |

**Task setting**   In this task, the goal is to minimize the cumulative training loss of a simple MLP (MNISTNET). All experimental setups (including model variants) and the LSTM-based optimizer baseline are borrowed from the open-sourced PyTorch implementation of L2LGD2[2] The default MNISTNET has one 20-dimensional hidden layers with Sigmoid activation. In addition, we also consider three other variants of MNISTNET: 1). MNISTNET-2Layer, which doubles the number of layers in MNISTNET; 2). MNISTNET-Big, which doubles the hidden layer dimension of MNIST-NET; 3). MNISTNET-ReLU, which replaces the Sigmoid activation in MNISTNET with ReLU. All models are trained for 1000 steps with a batch size of 128 on the MNIST dataset. We use a fixed 50/50 split of training and testing set for MNIST.

**Optimizer evaluation**   For each optimizer, the best learning rate is obtained from $\{0.0006, 0.001, 0.003, 0.006, 0.01, 0.03, 0.06, 0.1, 0.3, 1.0\}$. During the grid search, we train the network for 100 steps. After that, the network is retrained for 1000 steps with the best learning rate. We record the cumulative training loss and test accuracy of each optimizer for comparison. Note that following the L2LGD2 implementation, the grid search for the LSTM-based optimizer is applied at training time rather than test time.

**Search setting**   Again, we follow the search settings implemented in the L2LGD2 codebase for our experiment. The search is conducted on the default MNISTNET by training it for 100 steps on the training split. Standard early stopping procedure is enabled for this task. Empirically, we found that roughly 7.2% optimizers are terminated.

**Discovered optimizers**   We represent some of the discovered optimizer down below. Note that the forms of these optimizers are already simplified using the Sympy library.

$$\textbf{mnist1}:\ m_1 + RMSprop * exp(Adam)$$
$$\textbf{mnist2}:\ m_1 * (exp(Adam) + exp(exp(Adam))) + RMSprop$$
$$\textbf{mnist3}:\ m_1 + RMSprop * rd^{g^3}$$

Interestingly, the pattern $m_1 + RMSprop$ shows up quite frequently in the discovered optimizers for this task.

---

[2] https://github.com/chenwydj/learning-to-learn-by-gradient-descent-by-gradient-descent

**More experimental results**    In addition to the convergence figures in the main text, we also present the results in tabular form. Table 6 summarizes both the cumulative training loss and test accuracy of the optimizers on four MNISTNET models. All models are trained for 1000 steps. Our discovered optimizer achieves the best cumulative training loss and test accuracy for all cases.

### D.3    Image classification with ConvNet

**Task setting**    The goal of this task is to train a ConvNet on CIFAR-10 dataset. Following NOS-RL, the ConvNet has two 32-filter 3x3 convolution layers, each followed by ReLU activation and batch normalization [15]. We use a fixed held-out validation set of 5000 images for grid search. Note that the held-out validation set is used throughout the search phase, and it will be added back to the training set during final evaluation of the proposed optimizers.

**Optimizer evaluation**    The grid search is performed over $\{1e^{-5}, 1e^{-4}, 1e^{-3}, 1e^{-2}, 1e^{-1}, 1\}$ for 1 epoch of training, and the best learning rate is selected based on the accuracy on held-out validation set. After that, the optimizers are trained for a longer period of time (5 epochs). The batch size is set to 100.

**Search setting**    For this task, we disable early stopping (i.e. all optimizers will be counted into the budget.), to establish fair comparisons with NOS-RL's search budget; The reason is as follows: Although NOS-RL also aggressively early stops poor optimizers, the authors added them back when plotting Figure 4 in their paper; And since our estimation of NOS-RL's search budget comes from Figure 4, it would be rational to also disable it in our experiment. Each evaluation takes about 3 minutes to finish. And the duration for the entire search phase is around 7 hours on a single RTX 2080ti.

**Discovered optimizers**    Some of the discovered optimizers are shown below. Note that the forms of these optimizers are already simplified using Sympy library.

$$\textbf{conv1}: \ cd * drop_{0.1}(g)/ld$$
$$\textbf{conv2}: \ cd * sign(m_1) * |m_2|^{\sqrt{|ld|}/2}$$
$$\textbf{conv3}: \ drop_1(cd * m_1)$$
$$\textbf{conv4}: \ m_1 * (rd + |g|) * exp(cd)$$

**Further discussions on NOS-RL baseline**    NOS-RL applies Reinforcement Learning to train a LSTM controller to generate optimizer update rules according to a predefined pattern. Due to the training difficulty, NOS-RL adopts a multi-config-multi-run search strategy: It conducts the search multiple times with different subset of operators (unknown) and different optimizer length (5,10,15,20). Out of all search runs with different configurations, two families of best optimizers are reported in the paper. This leads to several challenges that prevent us from obtaining exact comparisons with NOS-RL: 1). It is difficult to know or measure the exact search cost of NOS-RL due to its multi-config-multi-run strategy. The paper only mentions that a single search run can finish in one-day with heavily parallelism on Google's infrastructures. Therefore, the best we can do is to make an estimation based on Figure 4 in their paper, where it shows that the controller begins to converge at least after 10k evaluations. 2). The particular subsets of operators used during each search run is also unknown; The paper only mentions that the search spaces generated by these subsets typically contains $10^6$ to $10^{11}$ update rules. As a result, we have to pick our own operator set to run the search on.

We conjecture that the main purpose of NOS-RL paper is to offer the discovered optimizer for practitioners to use, rather than providing a baseline to stimulate further developments of non-parametric optimizer search methods. This can be evidenced by its prohibitive search cost, and also by the fact that the source code is not released. The nature of NOS-RL, combined with aforementioned challenges, necessitate an open-sourced resource-friendly non-parametric optimizer search framework for the community, which we hope to provide in this work.

## D.4 Adversarial Attack

**Task setting** Adversarial attack aims at finding a norm-bounded perturbation to the input space that misleads the model predictions. In this case, the parameter to be optimized is the data itself. We use the AutoAttack library[3] to implement our experiments for this task. The library contains a set of defense methods, as well as an implementation of the APGD optimizer that we used as the baseline. The attack is conducted on the default test split of CIFAR-10 dataset, which contains 10000 images. Our metric of choice is attack success rate. Concretely, if the perturbed image successfully mislead the model's prediction into a wrong class, then the attack is successful for that image. The success rate is thus the percentage of images that the optimizer successfully attacked.

**Optimizer evaluation** The search is conducted on the Carmon2019 method. We use $\{0.001, 0.003, 0.006, 0.01, 0.03, 0.06, 0.1, 0.3, 1\}$ to search for the best learning rate. The grid search is conducted on only 1000 test images. After that, the optimizers will be evaluated by training for 100 steps.

**Search setting** During the search phase, all optimizers are evaluated with only 20 steps, as it is usually enough to identify top optimizers. To further reduce the search cost, we use only 400 images for grid search and 4800 (400 * 12) images for evaluation. The search takes around one GPU day to finish on a single RTX 2080ti.

**Discovered optimizers** We present some of the discovered optimizers for adversarial attack down below.

$$\textbf{attack1}: \ log(|cd + sign(g)|)$$
$$\textbf{attack2}: \ log(|cd + exp(g^3) * sign(g)|)$$
$$\textbf{attack3}: \ log(|ld + sign(RMSprop)|)$$

As mentioned in the main text, we found that many top optimizers are log-based. More specifically, these optimizers often have the form of $log(|decay + sign(\cdot)|)$. The discovered log-based optimizers are also highly effective when transferred to other defense models, as showing in Table 2.

**Discussion on Adaptive Projected Gradient Descent (APGD) optimizer** APGD is currently the strongest manually design and tuned optimizer for adversarial attack. It consists of two parts: 1). a momentum update rule and 2). a dynamic learning rate decay schema. The momentum update rule takes the following form:

$$z^{(k+1)} = Proj_{B_\epsilon^\infty(x_o)}(x^{(k)} + \gamma^{(k)} sign(\nabla_x L(x^{(k)})))) \tag{1}$$
$$x^{(k+1)} = Proj_{B_\epsilon^\infty(x_o)}(x^{(k)} + (1-\mu)(z^{(k+1)} - x^{(k)}) + \mu(x^{(k)} - x^{(k-1)})) \tag{2}$$

where $k$ is the current step and $\mu$ is the momentum ratio. The $\mu$ is tuned to be 0.25, much lower than the default momentum ratio for standard optimizers such as SGD and PGD. The dynamic learning rate decay schema halves the learning rate if a set of two conditions are satisfied at some predefined steps $\{w_j\}_{j=1}^{100}$:

$$\sum_{i=w_{j-1}}^{w_j-1} \mathbb{1}_{\mathcal{L}(x^{(i+1)}) < \mathcal{L}(x^{(i)})} < \rho * (w_j - w_{j-1}) \tag{3}$$

$$\gamma^{(w_j)} \equiv \gamma^{(w_{j-1})} \ \text{ and } \ \mathcal{L}_{min}^{(w_j)} \equiv \mathcal{L}_{min}^{(w_{j-1})} \tag{4}$$

where $\rho$ is a threshold term and $\mathcal{L}$ denotes the loss function. The steps $(w_j)$ to check for these conditions are set to $\{23, 42, 58, 71, 81, 88, 94, 100\}$. As we can see, APGD has a quite complicated form, and its design also packs a lot of human expertise. On the other hand, our automatically discovered optimizers are much simpler, while also rivaling the performance of APGD.

---

[3]https://github.com/fra31/auto-attack

### D.5 Node classification on graphs

**Task Setting**   We consider node classification task on graphs. The model of choice is Graph Attention Network (GAT). There exists many PyTorch implementations of GAT and its variants, each covers only some datasets. As a result, we have to use more than one codebase for this experiment. For training cluster-GAT on OGBN-Products dataset, we use the official implementation from OGBN library[4]. For training vanilla GAT on Cora dataset, we use pyGAT[5]. For training vanilla GAT on Citeseer, PubMed, and PPI dataset, we use the implementations from DGL library[6]. We follow the instructions provided in their README.md files to run all of our experiments. The only except is for Cora dataset, where we disable early stopping in the original implementation. The reason is that the default criteria often terminate training prematurely for our optimizers.

**Optimizer evaluation**   The grid search is conducted over $\{0.0003, 0.0006, 0.001, 0.003, 0.006, 0.01, 0.03, 0.06\}$, as we found that most of the optimizers' best learning rates (including Adam) fall into this range. After the grid search, all optimizers are evaluated under 4 random seeds. As mentioned above, all other hyperparameters, including the total number of epochs, batch size, weight decay, e.t.c., are set to their default values as in the original codebases.

**Search setting**   We deploy standard early stopping schema for this task. The percentage of early terminated optimizers is around 10% to 17% for vanilla GATs. We found that Cluster-GAT is much harder to optimize: roughly 25% of the optimizers are early terminated. The search is conducted on each dataset separately, and we will discuss the transferability of the discovered optimizers in the next paragraph. For Products dataset, we parallelize the search over 8 RTX 2080ti GPUs. All other datasets are ran on a single device. The search takes about 2 GPU days to finish for Products dataset, 1 GPU day for PPI and Cora, and 3 hours for PubMed and Citeseer.

**Discovered optimizers**   We present some of the discovered optimizers on each dataset down below. Interestingly, we found that sign-based optimizers dominate the graph learning task.

**products1**: $ld * sign(m_1) - Adam$

**products2**: $ld * (sign(m_1) - RMSprop)$

**cora**:  $sign(m_1) + m_3$

**citeseer**: $drop_{0.1}((sign(g) - m_1)/cd)$

**pubmed**: $sign(m_1) + sign(m_1 - drop_{0.1}(g^3))$

**ppi**: $drop_{0.1}(rd^2) * sign(m_1)$

**More experimental results on the transfer setting**   Among the optimizers listed above, we found that "product2" optimizer exhibits the highest level of transferability. As shown in Table 7, it outperforms Adam on all but the citeseer dataset. Note that for graph learning task, there exists a non-negligible performance gap between transfer and direct search settings. We conjecture that it is because different graph dataset might indeed require different optimizers. The reason is as follows: Most graph neural networks, including GATs, adopt the message passing framework, where the features of neighboring nodes are passed to the target node through their edges in the forward pass. Since the connectivity of nodes are defined by the adjacency matrix in the dataset, the computation graph (and thus the learning process) is inherently

Table 7: Performance of our discovered optimizers against Adam on GATs on five commonly used Graph datasets of diverse size. Results that use the same GAT implementations are grouped together.

| Dataset | Adam | products2 |
|---------|------|-----------|
| Products | $77.49\% \pm 0.56^{\dagger}$ | $\mathbf{79.98\% \pm 0.17}$ |
| Cora | $84.72\% \pm 0.32$ | $\mathbf{84.87\% \pm 0.29}$ |
| Citeseer | $71.70\% \pm 1.03$ | $\mathbf{71.78\% \pm 0.51}$ |
| PubMed | $\mathbf{78.20\% \pm 0.22}$ | $77.12\% \pm 0.53$ |
| PPI | $97.53\% \pm 0.45^{\ddagger}$ | $\mathbf{98.38\% \pm 0.07}^{\ddagger}$ |

[†] Our reproduced accuracy using ogbn-leaderboard's implementation is lower than the displayed number ($79.23\% \pm 0.78$).
[‡] F1 Score

---

[4] https://github.com/dmlc/dgl/tree/master/examples/pytorch/ogb/cluster-gat
[5] https://github.com/Diego999/pyGAT
[6] https://github.com/dmlc/dgl/tree/master/examples/pytorch/gat

encoded in the dataset itself. Moreover, some of the datasets and models we considered are inherently heterogeneous. For example, PPI is designed for inductive learning, whereas all other dataset are for transductive setting; and also the Cluster-GAT model we used for OGBN-Products dataset is inherently different from vanilla GATs.

### D.6 BERT fine-tunning on NLP datasets

**Task setting**   We use Hugging Face's official implementation of BERT finetuning task for our experiment[7]. The goal of this task is to finetune a pretrained BERT (base cased) model on a set of NLP datasets. Following the instructions on the official repo, we set the number of epochs to 5 for MRPC and WNLI, and 3 for CoLA, STS-B, and RTE dataset. We also observe that finetuning for more epochs generally harms the performance of all optimizers. The model is trained with a batch size of 32 on a single GPU. We refer the reader to Hugging Face's offical repo (link in the footnote) for more details on this task.

**Optimizer evaluation**   We use $\{2e^{-5}, 0.0001, 0.0003, 0.0006, 0.001, 0.003, 0.006, 0.01\}$ for learning rate grid search. Note that this grid is intentionally shifted to cover Hugging Face's default learning rate for AdamW ($2e^{-5}$). After the grid search, the optimizers are trained for 4 random seeds with the best learning rate.

**Search setting**   Similar to GAT task, our search is conducted on each dataset separately, and we will discuss the transferability of discovered optimizers later. We early stop optimizers if their performance (accuracy, Matthew's correlation, or Spearman's correlation) fall below the default threshold (0.2) during the grid search. Empirically, we found that roughly 13.5% optimizers are terminated. This rate is slightly higher than that of MNISTNET task (7.2%), because we are using the same threshold as MNISTNET task even though the accuracies (or correlations) on BERT fine-tuning tasks are much lower. For this task, we parallelize the search over 8 RTX A6000 GPUs. The search can be finished in less than 10 hours on all dataset.

**Discovered optimizers**   We present some of the discovered optimizers on each dataset down below. Similar to those found on the GAT tasks, many optimizers are sign-based. Note that the power operator in cola2 optimizer might return NaN, which will be mapped to 0 by the sign function.

$$\textbf{cola1}: \ drop_{0.1}(Adam + g^3)$$

$$\textbf{cola2}: \ sign(m_1^{clip_{0.003}(rd)+clip_{0.003}(sign(g))-sign(m_1)})$$

$$\textbf{mrpc}: \ drop_{0.1}(clip_{0.003}(sign(g) + sign(RMSprop) * rd))$$

$$\textbf{stsb}: \ sign(RMSprop + 2/(g + m_3))$$

$$\textbf{rte}: \ drop_{0.1}(clip_{0.003}(m_1 - \sqrt{|drop_{0.1}(g^3)|} * sign(m_1)))$$

$$\textbf{wnli}: \ sign(m_1) - m_3$$

**More experimental results on the transfer setting**
We found that both "cola2" and "rte" optimizer exhibits high level of transferability. Although they perform slightly worse than the optimizers directly searched on the target dataset, it still consistently outperforms AdamW by a sizable margin. The results are summarized in Table 8. Note that some of our reproduced results for AdamW is a bit different than the reported numbers on the Hugging Face repository. The reason is that we run each optimizer for 4 seeds and report the average results, whereas the official repository only records the number after a single run.

Table 8: Performance of our discovered "cola2" optimizer for BERT finetuning task. Results above baseline are bolded.

| Dataset | AdamW | cola2 optimizer | rte optimizer |
|---------|-------|-----------------|---------------|
| Cola | $59.56 \pm 2.04^\star$ | $\mathbf{60.05 \pm 2.38}^\star$ | $\mathbf{59.91 \pm 1.54}^\star$ |
| MRPC | $82.84 \pm 0.57^\ddagger$ | $\mathbf{85.48 \pm 0.74}^\ddagger$ | $\mathbf{85.60 \pm 0.68}^\ddagger$ |
| STS-B | $87.80 \pm 1.14^\dagger$ | $\mathbf{87.90 \pm 0.28}^\dagger$ | $\mathbf{87.91 \pm 0.98}^\dagger$ |
| RTE | $65.97 \pm 1.56^\ddagger$ | $\mathbf{66.52 \pm 1.83}^\ddagger$ | $\mathbf{68.50 \pm 1.93}^\ddagger$ |
| WNLI | $53.17 \pm 5.49^\ddagger$ | $\mathbf{56.34 \pm 0.00}^\ddagger$ | $\mathbf{55.28 \pm 1.83}^\ddagger$ |

$^\star$ Mathews Correlation.
$^\dagger$ Spearman Correlation.
$^\ddagger$ Accuracy (%).

---

[7] https://github.com/huggingface/transformers/tree/main/examples/pytorch/text-classification

### D.7 License

The Hugging Face libary we used is licensed under Apache License 2.0. All other public repositories are licensed under MIT License.