# OpenReview forum: "Efficient Non-Parametric Optimizer Search for Diverse Tasks"
_NeurIPS.cc/2022/Conference — NeurIPS 2022 Accept_

### Official Review · Reviewer_d4Bz · 2022-07-09

**Rating:** 7
**Confidence:** 4
**Soundness:** 4 excellent
**Presentation:** 4 excellent
**Contribution:** 3 good

**Summary:**

This paper designs a new search space over optimizers and a search algorithm combining rejection sampling with Monte Carlo search. They evaluate the optimizer search on image classification, adversarial attacks, training GNNs, and BERT fine-tuning.

**Questions:**

Notes:
- 38: generability -> generalization
- 137: m_1 is a gradient, not a state vector?
- 173: do you have a citation for this claim about NAS? also in regular hyperparameter optimization my experience is that many configurations are poor
- Figure 2: Side-by-side columns might be more clear than overlapping.

**Limitations:**

There was no real discussion of specific limitations in the “Conclusion and Limitations” section, only the acknowledgment that the paper does not solve everything.

**Strengths And Weaknesses:**

**Post-Rebuttal:** The authors answered my questions largely satisfactorily. Under the assumption that the new comparisons to regular HPO baselines and more careful discussion of the limitations of the search space (preferably in the main text for the latter if given extra space) are included in the revision I believe this paper should be accepted and have raised my score.

Summary:
This submission is cleanly formulated, fairly easy-to-read, and has a rather extensive evaluation. However, in the weaknesses I have listed concerns, especially about the automation, budgeting, and task selection, that make me hesitant to give it a stronger score.

Strengths:
1. The construction of the search space seems novel and improves upon past constructions like NOS-RL in terms of representation efficiency.
2. Code is released and extensive experimental details are provided.
3. The paper is clear and fairly straightforward to read, although terse in parts.
4. The optimization procedure is evaluated on a diverse set of deep learning tasks.

Weaknesses:
1. The search procedure comes with its own hyperparameters. Rejection sampling, score thresholding, and depth all require tuning and how the levels were chosen is not explained
2. The full pipeline is not truly automated, as for individual tasks the initial grid search for the step-size uses a different grid over learning rates. Given how cheap it is, it is unclear to me why not use the same one for all of them.
3. The overall search budget is somewhat opaque to me. In Section 4.1 the number of evaluations is fixed at 128. How much more expensive is this than just running SGD on a deep net on different tasks? Full-pipeline time comparisons would be very useful here. My rough estimation is that the proposed procedure takes 32x the amount of time to train a deep net. How does this compare if a regular parallel hyperparameter tuner such as ASHA (Li et al., 2018) is given the same budget but tunes only SGD/Adam?
4. It is unclear if the proposed search space could actually discover stateful optimizers such as Adam if they were not explicitly included. The paper uses Adam as an example frequently, but could the momentum terms actually be discovered by such a system from elementary operations (adds, sorts, etc.)?
5. A common difficulty with AutoML papers is guaranteeing that the results are indeed evaluated on a representative set of tasks, without arbitrary exclusions. For example, for the GLUE benchmark, why are only five of the nine tasks evaluated? For CNN training, why was only image classification looked at?

References:
1. Li, Jamieson, Rostamizadeh, Gonna, Ben-Tzur, Hardt, Recht, Talwalkar. A system for massively parallel hyperparameter tuning. MLSys 2020.

---

> ### Author Response · Authors · 2022-07-31
> **Response Part 3/3**
>
> Weakness 5: *"A common difficulty with AutoML papers is guaranteeing that the results are indeed evaluated on a representative set of tasks, without arbitrary exclusions. For example, for the GLUE benchmark, why are only five of the nine tasks evaluated? For CNN training, why was only image classification looked at?"*
>
> We agree that AutoML systems need to be evaluated on different tasks. In fact, this is exactly our motivation to initiate an extensive evaluation in this paper. While we are not able to exhaust all tasks, we try our best to include a diverse set of optimization tasks such as adversarial attacks, graph learning, NLP finetuning, and computer vision. It includes constraint, unconstraint, mini-batch, and full-batch tasks. This is not the case for prior optimizer search works. For example, NOS-RL only tests its search algorithm on the CNN classification task.
>
> The primary goal of the CNN experiment is to establish a direct comparison with NOS-RL. So we follow NOS-RL’s setting in this experiment, which only considers the standard classification task. We would be thrilled to see future users exploring the framework on more visual learning tasks.
>
> For GLUE, we use five tasks because they are much quicker to experiment on than the rest. Although our framework is ~100x faster than NOS-RL, it still requires 128 evaluations (we refer to limitations in the common reply for potential solutions). So given the limited resource, we prioritized task diversity over exhausting all datasets of a single task. We are happy to include more GLUE benchmarks given sufficient time and computing resource in the future, but it is challenging to include those more computational expensive GLUE tasks during the rebuttal.
>
> \
> Questions:
> 1. Thanks for pointing it out! We will change the wording.
> 2. You are right. We were trying to categorize them into “gradient-based” terms, perhaps this would be a more accurate description.
> 3. Yes, and we also have several pieces of supporting evidence for that:
>     - NASBench-201 [1] is a widely adopted benchmark for efficient NAS. This benchmark holds 15,625 architectures. 89% of them have accuracy above 80% on CIFAR-10, and only 2% have accuracy below 50%.
>     - NASbench-101 [2] is another larger earlier NAS benchmark. It consists of 423,624 architectures. 98% of them have accuracy above 80% on CIFAR-10, and only 0.5% have accuracy below 50%.
>     - Also, there is a common observation that random search produces strong results in most NAS search spaces (although this is only a piece of indirect supporting evidence) [3, 4]
> 3. Thank you for the suggestion. We agree. Plus, this will also make the bars thinner. We will modify the plot.
>
> Limitations: Thank you for pointing it out. Please refer to the common reply to all reviewers.
>
> \
> We sincerely hope our response addressed your concerns and questions. Please let us know if anything else comes up.
>
> \
> [1] Dong, et al. NAS-Bench-201: Extending the Scope of Reproducible Neural Architecture Search. ICLR 2020\
> [2] Ying, et al. NAS-Bench-101: Towards Reproducible Neural Architecture Search. ICML 2019\
> [3] Liu, et al. DARTS: Differentiable Architecture Search. ICLR 2019\
> [4] Li, et al. Random Search and Reproducibility for Neural Architecture Search. UAI 2020

---

> > ### Comment · Reviewer_d4Bz · 2022-08-06
> > **Note**
> >
> > Thank you for answering my questions; I have raised my score in my main review. Since it is a minor point I won't argue too much about search spaces in NAS having mostly good configurations, but I *wouldn't* use [3] and [4] as evidence for it. [3] is not random search at all and [4] is a very NAS-specialized version of it in which the different samples are highly entangled via network weights.

---

> > > ### Author Response · Authors · 2022-08-07
> > > **Reply to the score raise**
> > >
> > > Thank you for your patience in reviewing our response, and for raising the evaluation of the paper. We are very glad to hear that your questions and concerns are properly addressed. For the minor point on NAS search spaces, we completely agree with you that the 3rd point (paper [3,4]) in our response might not be used as evidence. We will base our argument solely on the statistics drawn from standard NAS benchmarks (our 1st and 2nd points in our reply to Question 3).

---

> ### Author Response · Authors · 2022-07-31
> **Response Part 2/3**
>
> Weakness 3: *"The overall search budget is somewhat opaque to me … ASHA (Li et al., 2018) is given the same budget but tunes only SGD/Adam?"*
>
> For our algorithm, we use 32 evaluations per traversal level and a total of 4 levels for all experiments (line 275). Evaluations within a single level are fully parallelizable. So if the user has access to 32 GPUs, the whole pipeline can potentially be finished by ~ 4 * a_single_run_time. But we ran most of our experiments sequentially due to resource limits in the lab environment.
>
> We report the duration of the entire pipeline in the Appendix for each task (e.g. line 728). But the numbers could be misleading: it depends not only on the search budget (num evaluations) but also on how well we optimize the searcher code. We have not optimized our searcher code aggressively, and there is great room for improvement. For example, the candidate sampler (including the proposed checkers) can be fully parallelized on the CPUs, making the searcher’s overhead negligible. Therefore, from here we define budget as the “total number of training steps of all candidates” to compare our method with ASHA.
>
> **Comparison with ASHA**
> - We argue that there are cases (tasks) where a poor optimizer with a near-optimal hyperparameter cannot outperform a better optimizer with a potentially suboptimal hyperparameter. For example, we consider the following setting: 1). MNISTNet task used in our ablation study (fastest task to run). 2). ASHA and our algorithm are provided with the same budgets. 3). we assume access to a single commercial server with 8 GPUs. In this case, both ASHA and our algorithm can be fully parallelized (i.e. fully leveraging all GPUs all the time). 4). We compare SGD/Adam tuned by ASHA with our discovered optimizer tuned by the default grid search.
> - For ASHA, we use $\eta= 2$ (halving) and a learning rate grid from 1e-4 to 1 (which covers the best learning rate for SGD and Adam). Instead of merging ASHA into the codebase, we compute how many candidates (learning rates) it can evaluate under the given budget and just evaluate all of them fully. This serves as the upper bound of ASHA’s performance.
> - From Table 6, our discovered optimizer produces 93.86%±0.23 test accuracy on MNISTNet, ASHA boosts the performance of SGD from 93.09%±0.17 to 93.09%±0.16 (the optimal lr happens to be quite close to an entry in our grid), and Adam from 91.54%±0.53 to 91.69%±0.45, but they are still outperformed by our discovered optimizer.
>
> Moreover, since the optimizer search problem is orthogonal to HPO, our framework could benefit from advanced HPO methods as they 1). improve the efficiency over grid search 2). avoid potentially devaluing optimizers because they are assigned with suboptimal learning rates.
>
> \
> Weakness 4: *"It is unclear if the proposed search space could actually discover stateful optimizers such as Adam if they were not explicitly included. The paper uses Adam as an example frequently, but could the momentum terms actually be discovered by such a system from elementary operations (adds, sorts, etc.)?"*
>
> This is a good question. Explicitly including Adam in our operator set is a practice borrowed from NOS-RL. Adam can also be discovered by combining two momentum terms (Figure 1).
>
> On the other hand, the current operator set does not include searching for momentum update rules (so is NOS-RL). But it is doable by adding operators with their own states. We tried this quickly before, but the discovered momentum rule performs similarly to regular momentums. Enabling the optimizer search framework to discover novel momentum terms is an interesting problem that is worth a separate work. This is the next direction we are actually looking into (Please also see the limitations in the common reply above).
>
> Moreover, although sometimes the discovered optimizer includes Adam, it is only a part of the update rule. The whole optimizer can outperform Adam consistently.

---

> ### Author Response · Authors · 2022-07-31
> **Response Part 1/3**
>
> Thank you for your positive and detailed feedback on the paper; we will address your concerns and questions down below.
>
> Weakness 1: *"The search procedure comes with its own hyperparameters. Rejection sampling, score thresholding, and depth all require tuning and how the levels were chosen is not explained"*
>
> We agree that the search process is not entirely free of hyper-parameters. However, it is much less than prior methods such as NOS-RL because training its RL component alone introduces many extra tunable parameters. Also, our search hyperparameters are held fixed throughout all our experiments, across all the tasks. We will discuss their choices further:
> 1. Score threshold: The purpose of score thresholding is mainly to reject apparently poor optimizers - such as those that lead to near-random-guess accuracy or exploding loss. So we simply set it (20%) to be slightly higher than random guessing accuracy (10% at maximum for our tasks) and used it for all our experiments.
> 2. We set and fix the number of total samples in Monte Carlo estimation to 32 - a multiple of 8 for parallelization (line 275). If this number is too small, then the MC scores for each branch would be of high variance; Setting this number too big does not hurt performance, but also might not be necessary as it consumes extra budget.
> 3. Depth: We choose ten because we found that most top optimizers are within this range (line 277). We also experiment with increased depth (15, 20) but only find marginal gains sometimes. The discovered longer optimizers are also less interpretable.
> 4. The number of traversal levels: We use a fixed number to fix the same budget for all experiments (recall that num_mc_samples * num_levels = total budget) (line 267). But in practice, this hyperparameter needs not to be preset or tuned; One can just progressively increase it and stop when there is no further gain in terms of the proposed optimizer's performance.
>
> Thank you for pointing it out. We will include further discussions on this matter in our paper.
>
> \
> Weakness 2: *"The full pipeline is not truly automated, as for individual tasks the initial grid search for the step-size uses a different grid over learning rates. Given how cheap it is, it is unclear to me why not use the same one for all of them."*
>
> We actually use one main grid. And we do NOT tune it just to get better accuracies. The reason that sometimes we report different grids is as follows:
> 1. We use [1e-5, 1e-4, 1e-3, 1e-2, 1e-1, 1] for ConvNet task proposed in NOS-RL. This grid is too sparse, but it is adopted in NOS-RL (Sec 4.3). So we have to use the same grid for a fair comparison with NOS-RL.
> 2. For all other tasks, we use this grid: [0.0001, 0.0003, 0.0006, 0.001, 0.003, 0.006, 0.01, 0.03, 0.06, 0.1, 0.3, 1.0]. One can always use the full grid. However, this grid is sometimes unnecessarily long for some tasks - the first or the last few rates are never selected. Due to the sheer size of experiments in this paper and limited resources, we often shrink this grid mildly to speed up the runs a little. But we faithfully report the truncated grid instead of the full grid for each task, which might leave a false impression that we are tuning it.
> 3. We always want to ensure that the grid covers the provided learning rate of our baseline optimizers for a fair comparison. For the BERT task, since the baseline (AdamW)’s learning rate is set to 0.00002 (outside this grid), we extend our grid to cover 2e-5 to avoid devaluing AdamW. (line 840, where there was a typo that we missed 0.0003)
>
> Also, we adopt grid search only for its simplicity, as our main focus is on the optimizer update rule. We conjecture that replacing it with advanced, efficient HPO algorithms will further improve the performance (later in Weakness 3).
>
> We will also add a standalone section to discuss this.

---

### Official Review · Reviewer_8MmY · 2022-07-12

**Rating:** 4
**Confidence:** 3
**Soundness:** 2 fair
**Presentation:** 1 poor
**Contribution:** 2 fair

**Summary:**

This paper presents an approach to learning to optimize (L2O) in which the goal is to automatically configure the best gradient update rule.

**Questions:**

- maybe I overlooked this, but do you have an example/reference that gives clear evidence for different tasks that have different best optimizers? Which datasets, for which learners, and which are the optimal optimizers, and what is the difference in performance/time?
- how are the L2O and essential optimization phases connected? Do you select a different update rule based on the MCT approach each time?
- what is generability?

**Ethics Review Area:**

["I don’t know"]

**Limitations:**

The authors do not discuss limitations. There is such a section and gladly the authors recognize that they have not solved the whole problem yet, but no concrete limitations are mentioned.

**Strengths And Weaknesses:**

The main strength of the paper is that it addresses a relevant problem and that it mostly discusses well the related work.

My major concern is the soundness and clarity of the paper. To me it did not become clear when the optimization happens? Is this an online procedure that updates the update rule over time? It does not become clear enough how the MCT approach is interleaved/connected to the essential optimization process. Without this, it is hard to judge on the rest, and I would make my final assessment dependent on how this is clarified.

---

> ### Author Response · Authors · 2022-07-31
> **Response Part 1/1**
>
> Correction on the summary: *"This paper presents an approach to learning to optimize (L2O)"*
>
> We respectfully point out that “an approach to learning to optimize” from the summary is inaccurate. In the introduction section, we differentiate our method (and NOS-RL) as a separate line of work for optimizer search than L2O (paragraphs 2, 3, 4). Concretely, L2O aims to find parametric update rules (line 33), where the update rule itself needs to be learned from data. On the other hand, our method and NOS-RL aim to find non-parametric optimizers without trainable parameters. You may also refer to Section 2.1, paragraph 2 for a re-emphasis on this distinction.
>
>  As mentioned in the paper, the update rule learned by L2O has poor generalization, preventing if rom being successfully used in real problems. In contrast, we show the proposed framework can  surpass existing optimizers on a diverse set of tasks.
>
> \
> Weakness: *"My major concern is the soundness and clarity of the paper. To me it did not become clear when the optimization happens? Is this an online procedure that updates the update rule over time? It does not become clear enough how the MCT approach is interleaved/connected to the essential optimization process."*
>
> Our algorithm is offline, i.e. we are not updating the optimizer on the fly. This is also true for one of our baselines: NOS-RL, which is closer to ours than L2O in terms of the paradigm. Given a task, our goal is to develop an efficient search algorithm to find a good non-parametric optimizer off-line. The search algorithm (MCT) does not contain learnable parameters itself, and need not be trained.
>
> Our framework follows a standard search pipeline:
> 1. Candidate proposal: the search algorithm generates an optimizer update rule from the search space.
> 2. Candidate evaluation: The update rule is evaluated from scratch by using it to optimize a model and use its accuracy to score the optimizer.
> 3. Search: The optimizer score is used to guide the search algorithm to propose new optimizers.
> 4. Loop: Repeat step 1 - 3 until a predefined search budget is reached.
> 5. Finally, the top 1 optimizers visited by the search algorithm will be returned.
>
> We hope this will help you understand the paper. If anything else comes up, please let us know.
>
> \
> Question 1: *"do you have an example/reference that gives clear evidence for different tasks that have different best optimizers?"*
>
> The Appendix (page 16 - 21) already includes a collection of top discovered optimizers for all tasks, including models and datasets. We also discussed and analyzed the noticeable difference in their forms for different tasks in the main text and Appendix. For example, the best optimizer for the adversarial attack is log-based (line:340 and line:769), whereas sign-based optimizers frequently emerge on attention models (line: 808, line: 851).
>
> The performance of the optimizers is reported in the experiment section and the Appendix. The runtime overheads of these update rules are similar to human-designed ones (such as Adam) as they both apply elementary operations on the gradient. But this is not the case for parametric optimizers like L2LGD2: It applies a neural network (LSTM) over the gradients, so it is typically slower.
>
> \
> Question 2: *"how are the L2O and essential optimization phases connected? Do you select a different update rule based on the MCT approach each time?"*
>
> We address this question in our response to the weakness section. Yes, the MCT approach samples a different optimizer each time.
>
> \
> Question 3: *"what is generability?"*
>
> We refer to whether the discovered optimizer can generalize to variants of a task (line 37-38), such as different training steps or model configurations.
>
> The proposed framework achieves the same level of generalization compared with its human design counterparts. This is not the case for parametric methods such as L2LGD2, which generalizes poorly to different training configurations or models (Figure 3)
>
> \
> Limitations: Thank you for pointing it out. Please refer to the common reply to all reviewers.

---

> > ### Comment · Reviewer_8MmY · 2022-08-08
> > **Answer to your Rebuttal**
> >
> > Dear authors,
> >
> > ok the routine now gets much clearer to me (and I wonder why the algorithm you wrote down in the rebuttal is not part of the paper, it should be).
> >
> > I am still not very convinced of the approach (in fact even of the problem as such). After having better understood the approach, I have even new concerns, which I am however open to discuss in case you can rebuttal them:
> >
> > * measuring an optimizer in terms of accuracy does not appear reasonable to me. You must account for convergence speed, so the measure should be some function of the learning curve. The metric you use here would have to be pointed out clearly.
> >
> > * the runtime of this optimization process must be counted towards the overall NAS runtime. You cannot do this offline, so for me in a sense it *is* an online procedure, only that you fix the update rule after some time (your budget).  It is my impression that the results section does not show these costs. Does the red curve (yours) in Figure 3 start with the optimizer chosen by the approach or is this search phase contained in the plot? I am inclined to believe in the first hypothesis, which would give your approach an unfair advantage.
> >
> > A general remark is that I wonder about whether there is any such thing like a single-best optimizer for all architectures. I would believe that the optimal update rule even strongly depends on the architecture itself. Of course, I would not expect such insights from your work, but if even the offline case is successful, then how much more we should expect from a kind of meta-learning approach that tries to predict the best update rule for a concrete candidate?
> >
> > I think the title of the paper is also too broad since it is really only about NAS.

---

> > > ### Author Response · Authors · 2022-08-09
> > > **Further discussion on your reply Part 2/2**
> > >
> > > \
> > > *"A general remark is that I wonder about whether there is any such thing like a single-best optimizer for all architectures."*
> > >
> > > Please correct us if we misinterpreted your question. Our understanding is that you think there is NO single-best optimizer for all scenarios (including different architectures)?
> > > We would like to emphasize that **we also DO NOT believe such one single best-optimizer for all cases exists either, as we discussed in the related work section (line 229).**  In fact, this argument is exactly the motivation of the entire optimizer search field, including our work. The rationale is that since there is no single-best optimizer that tops all other optimizers in every task (e.g. adversarial, GNN learning), we would want an automated framework to discover suitable optimizers for different tasks.
> > >
> > > We hope the above addresses your questions and concerns. Please feel free to let us know if any further questions arise. We are happy to discuss with you further.

---

> > > ### Author Response · Authors · 2022-08-09
> > > **Further discussion on your reply Part 1/2**
> > >
> > > Dear Reviewer 8MmY,
> > >
> > > Thank you for allowing us to elaborate further on your questions and concerns.
> > >
> > > Thank you for your suggestion. For the search procedure, Algorithm 1 in the appendix already describes it in detail. But we will add more comments to the pseudo-code, like those used in the rebuttal.
> > >
> > > \
> > > *"I think the title of the paper is also too broad since it is really only about NAS."*
> > >
> > > We'd like to address this first. Please correct us if we misinterpreted your response. We conjecture that you might misunderstand that we are searching optimizers for NAS (Neural Architecture Search). **We are NOT working on NAS or NAS-optimizer hybrid problem.** The paper is solely about how to search for optimizers efficiently. And in particular, we aim to search for non-parametric optimizers, as discussed in the Introduction section and also line 86-92. Hence we think the title is accurate, i.e. “Efficient Non-Parametric Optimizer Search for Diverse Tasks,” **where, by "diverse tasks" we meant the search method could apply to any generic task, such as image classification, adversarial attack, graph learning, BERT finetuning used in our paper.**
> > >
> > >
> > > \
> > > *"measuring an optimizer in terms of accuracy does not appear reasonable to me. You must account for convergence speed, so the measure should be some function of the learning curve. The metric you use here would have to be pointed out clearly."*
> > >
> > > We agree that optimizers can be measured by different metrics. We would like to point out that we did mention our specific metrics in the experimental sections (text such as line 289, and on the Table titles or footnotes such as Table 2 - 5), whether it is the cumulative training loss (e.g. line 289), attack success rate (e.g. Table 2 title), accuracy (e.g. Table 1), or correlations (e.g. Table 4 footnote).
> > >
> > > In this paper, we follow comparable prior art (NOS-RL) and mainly focus on final performance (e.g. NOS-RL exp 6.3 - 6.5). The final performance is obtained after training the model for a predefined epoch (same for all optimizers for a fair comparison). The reason is that there could be optimizers that converge fast but plateau quickly, which might not be ideal.
> > > However, we make the following remarks:
> > > 1. Our framework is generic and can cope with any user-defined supervision signals. For example, for a fair comparison with L2LGD2 on MNISTNet task, we use cumulative training loss to measure all optimizers, as stated in line 289. This measures the area under the learning curve when the training step is fixed.
> > > 2. Empirically, we found that the discovered optimizer generally converges at a similar speed as human-design optimizers (similar to Figure 3). We will plot more figures like Figure 3 for the tasks in our revision.
> > >
> > >
> > > \
> > > *"the runtime of this optimization process must be counted towards the overall NAS runtime."*
> > >
> > > We respectfully point out that we still think the process can be viewed as offline: Once the search is complete, the framework proposes a single best optimizer, whose format is no different from human-designed ones and can be used like a human-designed one as well - it can be transferred to different training settings (e.g. different model variants or datasets) within a single task (hence “generability”).
> > > In Figure 3, yes, the curve starts with the optimizer chosen by the approach. But we respectfully argue that it is reasonable for the following reasons:
> > > 1. Optimizers such as Adam also require a hand design process, which is typically lengthy and needs a lot of human expertise. After that, the hand-crafted optimizer (i.e. Adam) can be tested on different models and datasets. Our work mimics this whole process, but with the goal of automatizing the design part. In this sense, the search phase corresponds to the “hand-design phase” for Adam, which is also not included in Figure 3.
> > > 2. Empirically, our discovered optimizer exhibits **high level of transferability within each task** (i.e. across the model or dataset variants of the same task). This means that the user can search once and use the same discover optimizer for different variants of the task without searching again. For example, as discussed in lines 288-294, we only search once on subfigure 1’s model (and on only 10% of the steps); **The same discovered optimizer is directly transferred to all other subfigures (2-4)**. The results demonstrate the transferability of the discovered optimizer to variants of the MNISTNet task. For adversarial attacks, all different model variants in Table 2 are attacked using a single discover optimizer. The transferability experiment for GNN and BERT tasks can also be found in the appendix (Table 7 & 8).
> > >
> > > Moreover, the report and plotting protocol described above directly follow prior seminal work on optimizer search, such as L2LGD2 and NOS-RL, i.e., they also search for an optimizer and report the accuracy/curve of evaluating this optimizer by retraining a task from scratch.

---

### Official Review · Reviewer_deaa · 2022-07-13

**Rating:** 6
**Confidence:** 3
**Soundness:** 3 good
**Presentation:** 3 good
**Contribution:** 2 fair

**Summary:**

The paper tackles the problem of automatic optimizer search with a focus on deep learning problems.
The authors define a set of math operators, frame the problem as a tree-based path search and apply Monte-Carlo sampling to find an optimizer. Experiments on a diverse set of tasks shows that the proposed method outperforms a number of baselines.

**Questions:**

A few more questions (in addition to the novelty one from the previous section):

Line 224: How is the probing vector selected for different domains? Testing the equvalence of expressions based on only one vector might produce many false positives. How would it affect the perfromance of the method?

Line 278: What would be the authors guidlines on selecting the set of operators for a general task, if this set has to be adjusted from task to task? Does it mean that "On-the-fly constraint tree-traversal" and other heuristics will have to be adjusted as well?

Line 397: How is score thresholding defined from task to task? is the same threshold used for every task?

Figure 3 of experiments: Could the authors elaborate on the stability of found optimizers compared to other methods? For example, some error bars could help here, since the results are averaged only over 4 seeds.

**Limitations:**

-

**Strengths And Weaknesses:**

Post-rebuttal:

I would like to thank the authors for their effort answering my questions.
The authors satisfactory resolved most of my concerns, so I am happy to increase my score to 6.
_______________
The paper is well-written and easy to follow. The problem and solution are well-defined. The experiments are comprehensive and cover a large number of relevant tasks/problems.

My main concern is about the novelty of the work. The problem of automatically selecting an optimizer is not novel as outlined in the relevant work section. The closest competitor NOS-RL already frames the problem as a search on mathematical operators (and uses a similar set of basic operators). While I do agree that the proposed method is significantly more efficient than NOS-RL, it appears that the main novelty of the paper comparing to NOS-RL is plugging a more efficient data structure and applying Monte-Carlo on it (which as not a novel technique by itself). That makes the overall contribution of the paper quite incremental. I would appreciate if authors would elaborate a bit more on the novelty of their work.

---

> ### Author Response · Authors · 2022-07-31
> **Response Part 3/3**
>
> Question 1: *"Line 224: How is the probing vector selected for different domains? Testing the equivalence of expressions based on only one vector might produce many false positives. How would it affect the performance of the method?"*
>
> The probing vector is a pre-sampled random vector (dim = 50) from Gaussian distribution. It is the same procedure for different tasks since detecting mathematically equivalent forms is task-agnostic. Thanks for pointing it out! We will clarify this further in the revision.
>
> The false positive rate is actually negligible. We found that the results from probing match symbolic solvers 100% on 100 random samples. The reason is that optimizer update rules are element-wise operators, so feeding one random vector of dim=50 is equivalent to comparing the optimizers’ output under 50 inputs. This virtually eliminates the false positives.
>
> \
> Question 2: *"Line 278: What would be the authors guidelines on selecting the set of operators for a general task, if this set has to be adjusted from task to task? Does it mean that "On-the-fly constraint tree-traversal" and other heuristics will have to be adjusted as well?"*
>
> We acknowledge that for AutoML tasks, designing operator sets is often not free of human priors. However, our provided set of operators and heuristics are quite generic. Empirically, we found that the operator set is good enough for constructing high-performant optimizers for all tasks used in this paper, including MNIST, CNN training, BERT finetuning, GNN training, and adversarial attacks. And we use the same on-the-fly constraints, train-free test, and score threshold for all experiments.
>
> For future tasks, we recommend starting with the provided set and incorporating new operators when necessary. For inspiration on what to add, one might look into 1). what extra mathematical operations are used in existing optimizers for this task 2). the nature of the task (e.g. some might require max or sinusoid functions). Those are common practices for other AutoML domains as well.
>
> With an augmented operator set, other components in our algorithm can largely remain the same. 1). our train-free test, score thresholding, and math equivalent detection are generic and independent of the operator set 2). It could be beneficial to augment the on-the-fly constraints slightly when the new operators are added. It is because the on-the-fly constraints are mainly used for eliminating mathematical redundancies, and new redundancies might surface when new operators are included. But this should be an extra gain. We encourage the users to follow our general guidelines (line 211) for spotting these redundancies.
>
> Thank you for this good point; we will include the suggestions in the paper.
>
> \
> Question 3: *"Line 397: How is score thresholding defined from task to task? is the same threshold used for every task?"*
>
> It is the same for each task. Due to the space limit, we put the threshold used in the Appendix (C.1. first paragraph). Throughout our experiment, we used 20% for the accuracy metric and 10 for the loss metric.
>
> The purpose of score thresholding is mainly to reject apparently poor optimizers - such as those that lead to near-random-guess accuracy or exploding loss. So we simply set it (20%) to be a bit higher than random guessing accuracy (10% at maximum for our tasks) and used the same number for all our experiments.
>
> \
> Question 4: *"Figure 3 of experiments: Could the authors elaborate on the stability of found optimizers compared to other methods? For example, some error bars could help here, since the results are averaged only over 4 seeds."*
>
> We disabled the error bars in Figure 3 to make it look cleaner. The error bars for this experiment are reported in Table 6 in the Appendix (page 17). The average error bar on accuracy over four model variants is 0.2675 for our optimizer, similar to SGD (0.265) and better than all other optimizers (e.g. Adam: 0.53, l2lgd2: 0.5625)

---

> > ### Comment · Reviewer_deaa · 2022-08-09
> > **Increase my score to 6**
> >
> > I would like to thank the authors for their effort answering my questions.
> > The authors satisfactory resolved most of my concerns, so I am happy to increase my score to 6.

---

> > > ### Author Response · Authors · 2022-08-09
> > > **Reply to score raise**
> > >
> > > We thank the reviewer for going through the response and for raising the evaluation. We are very glad that your questions and concerns are properly addressed. If any further questions come up, please feel free to let us know, and we are more than happy to discuss them with you.

---

> ### Author Response · Authors · 2022-07-31
> **Response Part 2/3**
>
> **Our technical contributions:**
>
> **Instead of formulating it as a bottom-up sequential prediction task, we frame it as a top-down tree search.** This requires a re-design of the framework from the ground-up, including 1). search space arrangement 2). search algorithm 3). A set of techniques for improving the performance (accuracy or efficiency).
> 1. Our search space is inspired by viewing optimizer update rules as math formulas. We leverage their innate tree structure to build a super-tree, where each branch represents an optimizer. Because of this super-tree space, we can adapt tree traversal algorithms to the optimizer search task, which has not been done before. Note that the search space differs from the operator set. Those math operators are not new (line 139-140) - they are used in all hand-designed optimizers. So it is straightforward to choose them. What’s important is how to arrange them into a search space (super tree), as this depends on how we frame the problem. The search space arrangement set the tone for the entire framework.
> 2. Our choice of the search algorithm is a direct result of the above arrangement (super-tree). Moreover, we do not directly apply MC sampling to the optimizer search. As discussed in Sec 2.2, we experimented with MCT for optimizer search problem but found that it can not produce satisfying results because 1). the search space has a large portion of poor optimizers that makes random unrolling sample inefficient. 2). there also exist mathematical redundancies that make searching for optimizers inefficient. These problems arise from the optimizer search task specifically and are mostly unique challenges within our framework. We propose and ablate a set of techniques that solves those problems, including train-free test, score thresholding, on-the-fly constraint traversal, and equivalent optimizer detection. These techniques are an integrated part of our framework.
> 3. Our contributions can also be seen by drawing connections to DARTS, the founding paper of Efficient NAS [1]. The key innovation in DARTS is to rearrange the search space into a Super Network, enabling simple pruning algorithms to be used as the search algorithm. The operator set of DARTS - e.g. Conv and Pooling - is not new either, as they are also used in handcrafted architectures and prior NAS works. But how to arrange them matters most.
>
> To summarize, the paper introduces an integrated framework that differs from the prior anchor in formulating the problem, leading to fundamental distinctions in search space, search algorithm, and techniques for further boosting efficiency. The differences are non-trivial and parts of a unique entity. We hope our response could change your view of the paper. If you have any further questions on this, please let us know.
>
> [1] Liu et al. DARTS: Differentiable Architecture Search. ICLR 2019\
> [2] Andrychowicz et al. Learning to learn by gradient descent by gradient descent. NeurIPS 2016\
> [3] Zoph and Quoc. Neural Architecture Search with Reinforcement Learning. ICLR 2017
>
> *END OF FIRST RESPONSE*

---

> ### Author Response · Authors · 2022-07-31
> **Response Part 1/3**
>
> Weakness: *"My main concern is about the novelty of the work … I would appreciate if authors would elaborate a bit more on the novelty of their work"*
>
> Thank you for allowing us to elaborate on our work's novelty. We will explain it in three parts
>
> **Impact of our work: why is it important to make optimizer search efficient:**
>
> 1. A framework that can automatically find the best optimizer for each machine learning task will be revolutionary, which will not only lead to higher performance than using Adam/SGD for every task, but also save a significant amount of human effort in optimizer design. NOS-RL was proposed 5 years ago to address this problem. However, it is not used in practice or studied by other research groups due to the efficiency issue: It takes over 10,000 evaluations to discover a good optimizer. This amount of computation is prohibitive for anyone without access to Google’s infrastructure.
> 2. Our proposed framework enables optimizer search in 128 evaluations, making optimizer search applicable to many real-world tasks. Our framework is efficient, scalable, and generalizable - criteria essential for democratizing research and applications in optimizer search. While optimizer search is not a new research field (established before NOS-RL) [2], there hasn’t been a framework that checks all three boxes, which makes optimizer search truly applicable to tasks beyond image classification problems.
> 3. Empirically, we demonstrate that the proposed efficient framework enables finding state-of-the-art solvers for each particular task, including image classification, graph neural network training, adversarial attack, and BERT finetuning, even with academic-lab-level resources.
>
> **It is non-trivial to make NOS-RL efficient:**
>
> NOS-RL’s limitation in efficiency comes from how it frames the problem as a whole, and it is difficult to drastically improve it by swapping its components or making incremental changes.
> 1. **NOS-RL frames the problem as a bottom-up sequential prediction task (line 144).** Its pipeline is directly borrowed from RL-based NAS [3]: it applies RL with an LSTM controller to output an optimizer. Because it consumes a lot of samples to train an RL agent, it restricts its search space to a simple predefined pattern (line 144), just like its NAS counterpart. Those predefined patterns are not the ideal representation for optimizers as we explained in line 141-155. So NOS-RL is an integrated framework on its own, where the search space and algorithm designs are interdependent.
> 2. The design of the NOS-RL framework inherently limits its efficiency. This was the same story of RL-based Neural Architecture Search (NAS), which is why more efficient frameworks like DARTS are proposed later (more on this later). The prohibitive search cost (100x than ours) makes applying to different tasks infeasible. For this reason, it needs google’s advanced infrastructure to search on even a small ConvNet task. Because of this, the code cannot be released, according to the author. As a result, people are not utilizing or following their framework since its debut five years ago.
>
> *TO BE CONTINUED*

---

### Author Response · Authors · 2022-07-31
**Common reply to all reviewers**

**Summary**\
We thank our reviewers for their valuable feedback. We appreciate that the reviewers commented positively on the presentation (R1 & R3), soundness (R1 & R3), novelty (R3), relevance (R1 & R2 & R3), and the fact that our proposed method is significantly more efficient than comparable methods under extensive evaluations that cover a large number of relevant tasks (R1 & R3). We’d also like to acknowledge the insightful questions and concerns that our reviewers raised and that the reviewers explicitly granted us an opportunity to elaborate on these points. We will address them in our responses and incorporate them into our paper.

**Limitations**\
Here, we address a common concern raised by R2 and R3 on the limitations of the work.
We agree that we did not explicitly bullet-pointing the limitations in the paper. Rather, as pointed out by our reviewers, we state that our view of this work is a starting point for an efficient, scalable, and generalizable framework for optimizer search. And we expect plenty of room for improvement for future works. Through our continuing exploration of the framework post submission and the insights provided by our reviewers, we identify the following concrete limitations of the method:
1. We use precomputed momentum terms as input to our search space. This is a practice we borrowed from NOS-RL. Adding commonly used terms eases the job of the search algorithm because it does not have to rediscover them from scratch every time. However, searching for novel momentum update rules could potentially help find even stronger optimizers. In principle, our framework allows it: one can do this by inserting an operator with its own internal state. This is in fact one of the next directions we are looking into.
2. Identifying proper hyperparameters for an optimizer is essential for evaluation. In the current work, we use a simple grid search to discover the best learning rate for an optimizer. While it works fine for our tasks, this could potentially be suboptimal as it might underestimate some optimizers. Leveraging advanced fast HPO during the search phase would be another direction to explore.
3. Although our framework is 100x faster than the comparable method (NOS-RL, which requires google’s highly specialized infrastructure to run), it still requires 128 evaluations in the search phase. These evaluations can be largely parallelized. But potentially, the efficiency can be improved further with better search algorithms, more train-free tests (we proposed one in our work), knowledge transfer, e.t.c.

We have added a discussion on these limitations in our paper (Appendix E, marked BLUE).

---

### Author Response · Authors · 2022-08-08
**To all reviewers.**

We thank all reviewers again for your efforts in reviewing our paper.

We are glad to see that we addressed Reviewer d4Bz's questions during the rebuttal, who **then raised the score to 7:Accept**.

Since we haven't heard back from all reviewers, we would like to reach out to see if our rebuttal response has addressed your questions and concerns?
As the discussion period is close to an end, if you have any further questions, we are more than happy to discuss them further.

Please kindly let us know your feedback. Thank you for your time and help!

Best,\
Paper6951 Authors

---

### Meta-Review · Area_Chair_Kids · 2022-08-21

**Recommendation:** Accept
**Confidence:** Certain

**Metareview:**

Two of the reviewers were positive on this paper and thought the work would be of interest to the community.  One reviewer felt the paper lacked clarity, but the other reviewers disagreed.  I think there will be interest in the optimizer search for deep learning that this paper presents and feel it should appear in NeurIPS.

**Award:**

No

---

### Decision · Program_Chairs · 2022-09-14

Accept